# Extensive topographic remapping and functional sharpening in the adult rat visual pathway upon first visual experience

Joana Carvalho ●*, Francisca F. Fernandes, Noam Shemesh ●*

Laboratory of Preclinical MRI, Champalimaud Research, Champalimaud Centre for the Unknown, Lisbon, Portugal

* Joana.Carvalho@research.fchampalimaud.org (JC); Noam.Shemesh@neuro.fchampalimaud.org (NS)

## Abstract

Understanding the dynamics of stability/plasticity balances during adulthood is pivotal for learning, disease, and recovery from injury. However, the brain-wide topography of sensory remapping remains unknown. Here, using a first-of-its-kind setup for delivering patterned visual stimuli in a rodent magnetic resonance imaging (MRI) scanner, coupled with biologically inspired computational models, we noninvasively mapped brain-wide properties—receptive fields (RFs) and spatial frequency (SF) tuning curves—that were insofar only available from invasive electrophysiology or optical imaging. We then tracked the RF dynamics in the chronic visual deprivation model (VDM) of plasticity and found that light exposure progressively promoted a large-scale topographic remapping in adult rats. Upon light exposure, the initially unspecialized visual pathway progressively evidenced sharpened RFs (smaller and more spatially selective) and enhanced SF tuning curves. Our findings reveal that visual experience following VDM reshapes both structure and function of the visual system and shifts the stability/plasticity balance in adults.

**Data Availability Statement:** The raw data supporting the conclusions of this article is publicly available at the Open Neuro repository; doi:10. 18112/openneuro.ds004509.v1.0.0. The code for

## 1. Introduction

During critical periods of development, neural circuits undergo massive plasticity and organization processes that are strongly shaped by sensory experience [1,2]. At later stages of life, these plastic changes must reach a certain degree of stability to ensure that the gained functional refinements persist over time. Understanding the dynamics of stability/plasticity balances and how they are sculpted by experience is pivotal both for identifying mechanisms underlying normal/aberrant development and for recovery from injury.

Most studies demonstrating plasticity in rodents have focused on local features. For example, seminal electrophysiological and calcium recordings studies revealed that activity in specific junctions of the rodent visual pathway becomes highly refined during the first approximately 4 to 5 weeks of life [3,4]. The initially broadly tuned cortical neurons specialize towards well-defined functional properties, i.e., sharper spatial frequency (SF) tuning, and an orderly cortical arrangement of visual areas emerges such that neighboring neurons respond to nearby positions in the visual field (retinotopic organization) [5,6]. Visual experience refines

the preprocessing pipeline, the Micro Probing mapping of RFs and visual field reconstruction is available for free download at https://github.com/Joana-Carvalho.

**Funding:** This study was funded by the European Research Council (ERC) (agreement No. 679058 awarded to NS), as well as by the European Union's Horizon 2020 research and innovation programme under the Marie Sklodowska-Curie grant agreement No. 101032056, awarded to JC. The authors acknowledge the vivarium of the Champalimaud Centre for the Unknown, a facility of CONGENTO which is a research infrastructure co-financed by Lisboa Regional Operational Programme (Lisboa 2020), under the PORTUGAL 2020 Partnership Agreement through the European Regional Development Fund (ERDF) and Fundação para a Ciência e Tecnologia (Portugal), project LISBOA-01-0145-FEDER-022170. The funders had no role in study design, data collection and analysis, decision to publish, or preparation of the manuscript.

**Competing interests:** The authors have declared that no competing interests exist.

**Abbreviations:** AC, auditory cortex; BG, basal ganglia; fMRI, functional magnetic resonance imaging; GE-EPI, gradient-echo echo-planar imaging; HC, healthy control; HRF, hemodynamic response function; LGN, lateral geniculate nucleus; MC, motor cortex; MRI, magnetic resonance imaging; pRF, population receptive field; PSC, percentage signal change; RF, receptive field; ROI, region of interest; SC, superior colliculus; SF, spatial frequency; VC, visual cortex; VD, visual deprivation; VDM, visual deprivation model; VE, variance explained.

immature receptive fields (RFs) and the underlying neural connectivity, initially established by spontaneous activity, to improve their selectivity [7,8]. As the critical period ends (beginning of the sixth week), the plastic potential of the brain decreases and gradually reaches a stable state to support network stability [4,9].

Despite the importance of plasticity, for example, disease and recovery from injury in adulthood, whether a large-scale topographic remapping could be achieved during adulthood remains an open and controversial question. For example, in the visual deprivation model (VDM), [4,10,11], where animals are reared in darkness from birth and before exposure to light, the system is in an aberrant state on multiple scales. From a cellular perspective, RFs do not exhibit the sharp properties of normal rodents, and cortical function and structure resemble the conditions typically observed before eye opening (P14). In particular, broad spatial frequency tuning selectivity, [4,12,13] are found. Furthermore, at the population level, a lack of orderly visual maps and imprecise RF tuning [8,14] were found. Behaviorally, visual acuity remains low [15,16].

Exposure to light following VDM provides an outstanding opportunity to investigate plasticity/stability balances in adulthood. Whether first light exposure in adulthood promotes a normal development of the visual system is highly controversial, with some electrophysiology studies suggesting that VDM results in permanent deficits most likely driven by aberrant excitatory–inhibitory balances [13,17], supporting the view of a remarkable degree of stability in the adult visual system [11,18]. Others however, support the view that the brain remains plastic after the critical period by cortical remapping, in particular, through rescaling and displacing RFs, in response to visual deprivation [19,20]. These findings are corroborated also by lesion studies [19,20]. Similar mechanisms of plasticity have been also reported in monocular deprivation. The unilateral eye closure for a brief period of time (days) during the critical period decreases the responsiveness of cells in primary visual cortex (V1) to the deprived eye and it induces a shift in ocular dominance of binocular neurons towards the eye that receives visual input [3,11,21]. The duration of the monocular deprivation period, extent and age-dependence of ocular dominance plasticity in adulthood is also controversial. While some studies report extensive plastic changes in adult animals following brief periods of monocular deprivation up to 7 days [22–24], others fail to detect it in mature animals [9,25,26].

Although the studies above provide some evidence that a reorganization occurs upon first light exposure in adulthood, it is still unclear whether the system itself exhibits a convergence to the normal topographical mapping and whether it does so in unison or whether different areas of the brain have distinct dynamics. This gap stems from the macroscopic nature of entire networks and the inherently multidimensional time scales involved. Insofar measurements of RF properties in rodents have been limited to electrophysiological measurements of few sparsely distributed neurons [4,19,20] and by calcium imaging [27] focusing only on isolated brain regions, lacking the pathway-wide perspective [28,29]. Furthermore, RF properties are typically compared pre- and post- lesion/conditioning [19,20,29]. The difficulty of monitoring the same cells (e.g., via electrophysiology or calcium imaging) at different time points may lead to neuronal changes unrelated to plasticity. In addition, these methods are invasive which may introduce confoundings in the assessment of plasticity. These bottlenecks limit our knowledge on how the entire pathway adapts and generates a specialization of detailed RFs and SF, which is critical for developing future therapies and rehabilitation strategies.

Here, to enable the investigation of entire-pathway plasticity, we bridge this critical gap using preclinical high-field functional magnetic resonance imaging (fMRI) coupled with a novel setup capable of delivering complex visual stimuli in the scanner (Fig 1). MRI provides the required whole-pathway view and longitudinal capacity, and population receptive fields (pRFs) properties are routinely measured noninvasively in humans and nonhuman primates,

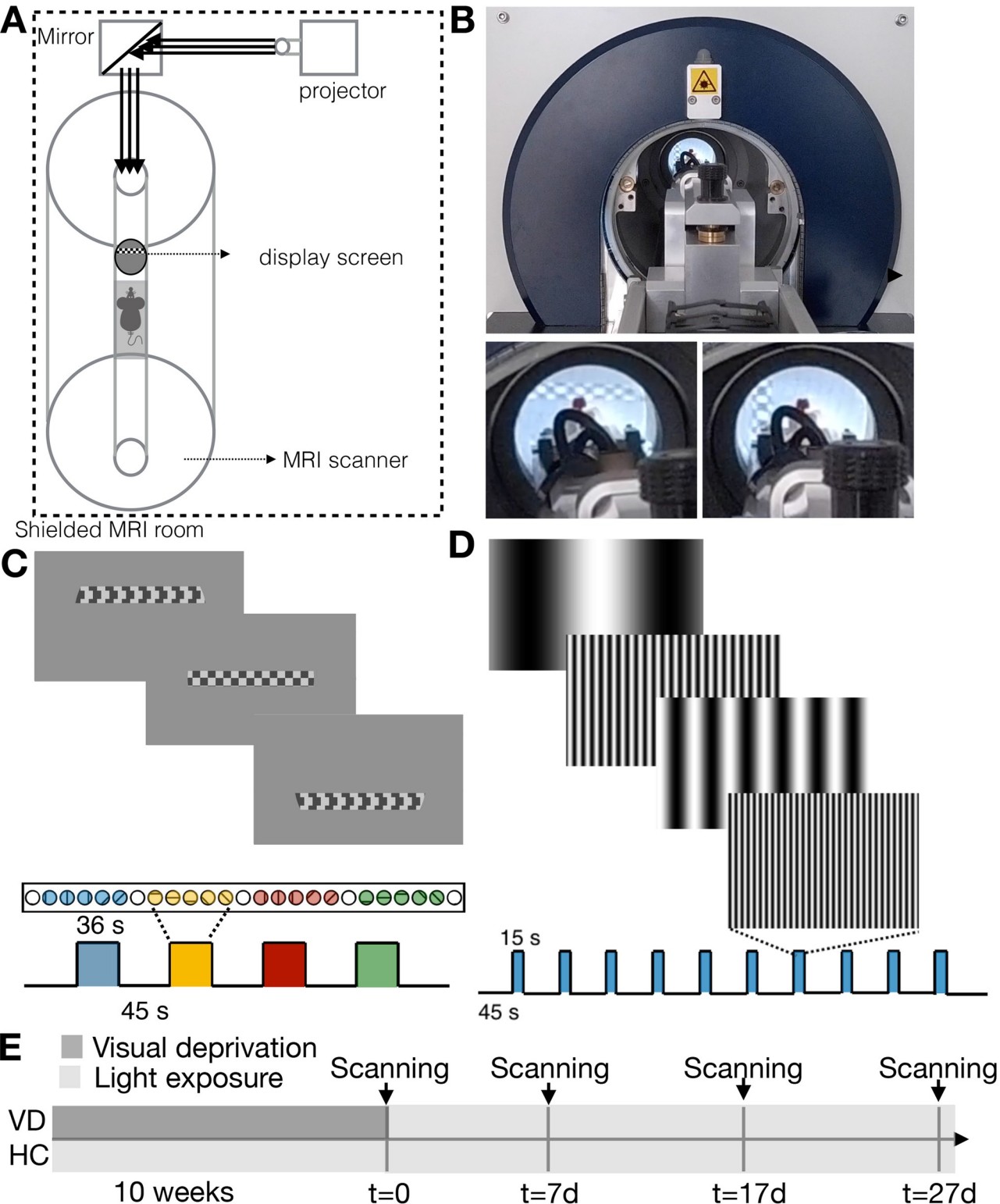

**Fig 1. The complex visual stimuli setup for preclinical MRI scanners, stimulation paradigms, and scheme of the dark rearing timeline.** (A) Visual stimulus display setup. (B) Picture of the visual stimulus displayed inside the scanner. (C) The retinotopy stimulation paradigm: checkerboard bar moving in 8 different directions (2 directions per stimulation block during 36 s followed by a 45 s rest period, repeated 4 times). (D) The SF tuning paradigm: 15 s stimulation period followed by a 45 s baseline. Ten different SFs were randomly presented at each stimulation block ranging between 0.003 and 0.5 cpd. (E) Timeline of the visual deprivation experiment for HCs and visually deprived (VD) animals. HC, healthy control; MRI, magnetic resonance imaging; SF, spatial frequency; VD, visual deprivation.

although at a coarse level of detail [30,31]. In addition, high-field preclinical MRI has proven ability to map in a highly detailed manner brain-wide plasticity [32] after sensory deprivation [33], peripheral nerve injury [34], and stroke [35]. However, delivery of such complex stimuli for rodents, which would enable a broad spectrum of experiments that are not possible to perform in human or primate counterparts, was insofar considered "impossible" due to space constraints in preclinical scanners. Using this first-of-its-kind setup, we mapped in detail the topographical and neuroanatomical organization of the entire rodent visual pathway for the first time, thereby linking the population level RFs vis-a-vis electrophysiology of entire visual areas at a whole pathway-level and in a noninvasive manner. We could then probe the plasticity/stability balance in an animal model where control can be exerted on the visual landscape and rearing conditions, i.e., adult rodents using the VDM (VD animals were born and kept in the dark until they were first exposed to light during the first MRI scan; husbandry and animal preparation for the MRI scanners were performed using red light), and follow if and how the system specializes in terms of RFs and SF tuning curves. Our results suggest that light exposure in adulthood results in an extensive topographical remapping and functional sharpening. The outcomes of this study have important implications for visual rehabilitation and restoration therapies.

## 2. Results

### 2.1. Retinotopic organization of the rat visual pathway mapped via fMRI

Fig 4 and Figs B, C and D in S1 Text show that the complex visual stimuli setup elicited reliable, temporally reproducible, and robust BOLD activation throughout the entire visual pathway in response to both retinotopic and spatial frequency tuning stimuli.

To validate the relevance of the complex visual stimulus setup, we first set to perform retinotopy in the rat using fMRI. The RF properties of the population of neurons within each voxel—referred to as "population RF" (pRF) [30]—were mapped voxelwise. Each pRF was modeled by a 2D Gaussian model and therefore characterized by a center (eccentricity and polar angle-phase) and a size. In addition, we mapped the pRF profiles, which capture the visual representation of each pRF through visual field sampling using very small (0.01˚) probes [36]. Fig 2A shows retinotopic (phase) maps averaged across animals for 3 representative slices containing the main junctions of the visual pathway, namely, lateral geniculate nucleus (LGN), superior colliculus (SC), and visual cortex (VC) areas. For all the studied areas, clear retinotopic organization was evidenced, and neighboring voxels responded to adjacent positions in the visual field as expected. Note that the visual information crosses at the optic chiasm, so the pRFs in the left hemisphere respond to the visual information presented on the right part of the visual field and vice versa (Fig 2B). In VC, the phase variation occurs along each cortical layer and it does not appear to vary across cortical depth. In the SC and LGN, strongly organized retinotopic maps were also observed.

The visual field representation for a determined region of interest (ROI) can be reconstructed by: (1) converting the pRF profiles into heat maps (histograms of a 30 × 30 bin grid weighted by its bin variance explained); and (2) aggregating the RFs across the voxels in an entire ROI via a normalized sum [37,38]. Fig 2C shows the visual field reconstruction at the level of VC, corresponding to what the animal is actually seeing. The yellow regions of the visual field were more highly sampled by the VC during the retinotopic paradigm. The overlap between VC visual field reconstruction and the available field of view is evident and, due to animal bed constraints, corresponds only to the top half of the visual field.

Using our setup, we could also determine the size of the RFs. In the VC, RF size changes across cortical layers and is constant within layers (Fig 2D), as expected. Fig 2E shows how

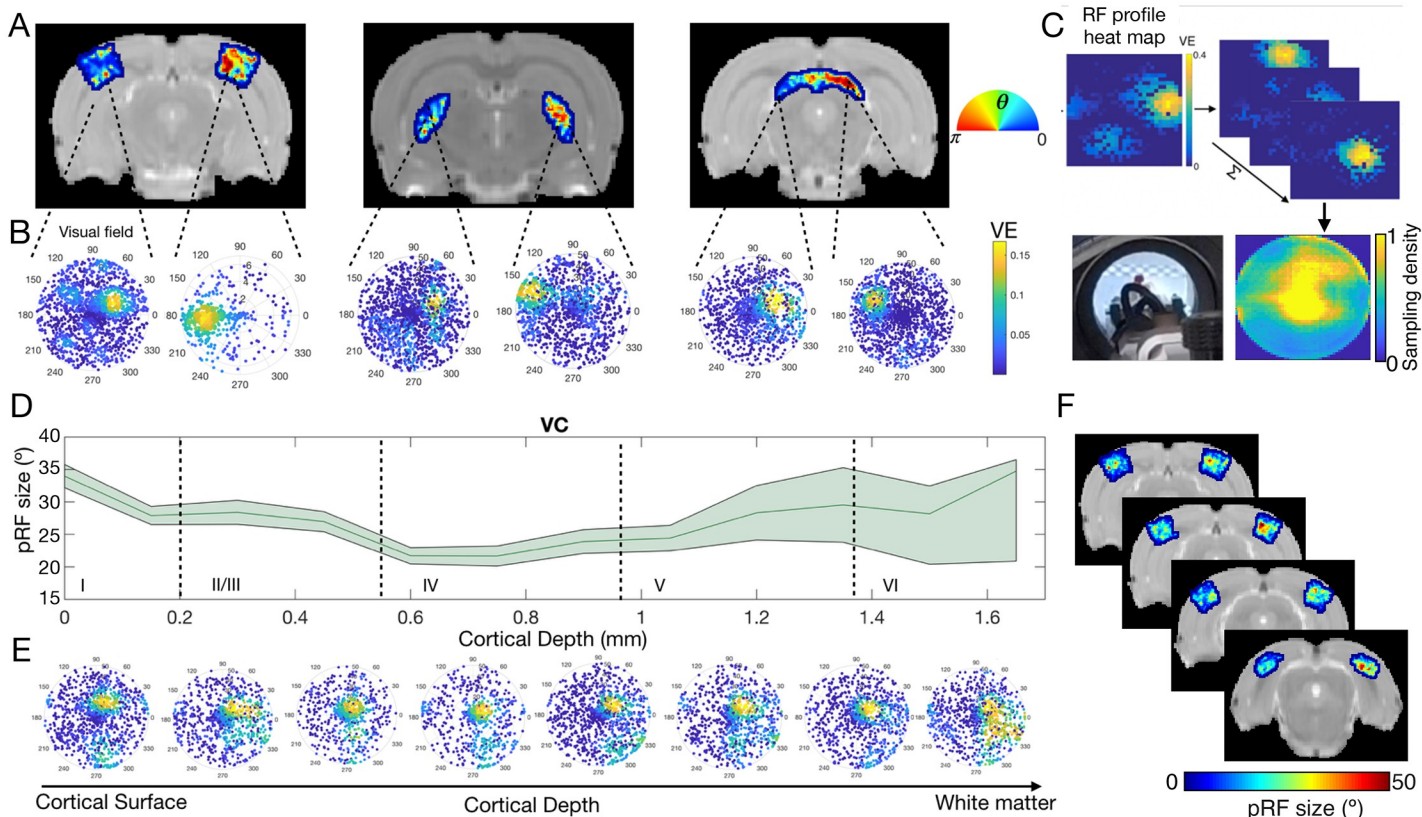

**Fig 2. PRF estimates of HC animals across ROIs and cortical layers at t = 0.** (A) Phase maps averaged across animals obtained for VC, LGN and SC, respectively. The color bar shows the preferred angle estimated per each voxel. (B) Visual representation of 2 pRF profiles located in the left and right hemispheres, respectively. The color bar shows the VE of each individual probe. (C) Average visual field reconstruction maps across animals for VC (obtained by summing the RF maps across some voxels of VC) and an image of the visual setup depicting the portion of the field of view covered by the animal bed. (D) Profile of the pRF size across cortical depth averaged across subjects, obtained from 2 slices of the VC. The green area corresponds to the 10% confidence interval. (E) Visual representation of 8 pRF profiles located across layers of the VC. (F) pRF size maps averaged across animals in 4 different slices of the VC. The color bar corresponds to the degree of visual angle. The data underlying this figure can be found here: doi:10.18112/openneuro.ds004509.v1.0.0. HC, healthy control; LGN, lateral geniculate nucleus; pRF, population receptive field; RF, receptive field; ROI, region of interest; SC, superior colliculus; VC, visual cortex; VE, variance explained.

pRF sizes vary with cortical depth. While superficial and deeper layers present larger pRFs, layer IV contains the smallest pRFs (Fig 2E). This is also clearly depicted in the pRF profiles across cortical layers (Fig 2F). These findings are in line with electrophysiological studies in cats, and with human fMRI reports where the smallest RF sizes are found in layer IV and the largest at layer VI [39,40]. The uncertainty associated with the pRF size estimates increases with cortical depth, likely due to the depth profile of our surface coil and the choice of coronal slices, which impart more partial volume effects in the slice direction.

Importantly, to ensure that our pRF estimates are reliable, we have calculated the pRFs in control areas that are not believed to be involved in processing visual stimuli, in particular, in the auditory cortex (AC), motor cortex (MC), and basal ganglia (BG) for the healthy controls (HCs) and VD animals for t = 0. As expected, the variance explained (VE) in visual areas is much higher than the VE in nonvisual areas (approximately 0.03), Fig H in S1 Text. Note that we have set a threshold for VE so that we only retain the pRFs whose variance explained is above 0.05, thus excluding very noisy voxels. Thus, our results reflect VE much higher than the variance explained of "pRFs" in nonvisual areas. The number of voxels excluded can be found in Table E in S1 Text.

## 2.2. Spatial frequency selectivity across the rat visual pathway mapped via fMRI

Spatial frequency selectivity is another core characteristic of the mammalian visual system, and its whole-pathway features in rodents have never been measured. Here, we measured spatial frequency selectivity across multiple structures of the rat visual pathway and derived their specific SF tuning curves (Fig 3). Fig 3A shows the definition of the ROIs (LGN, SC, and VC)

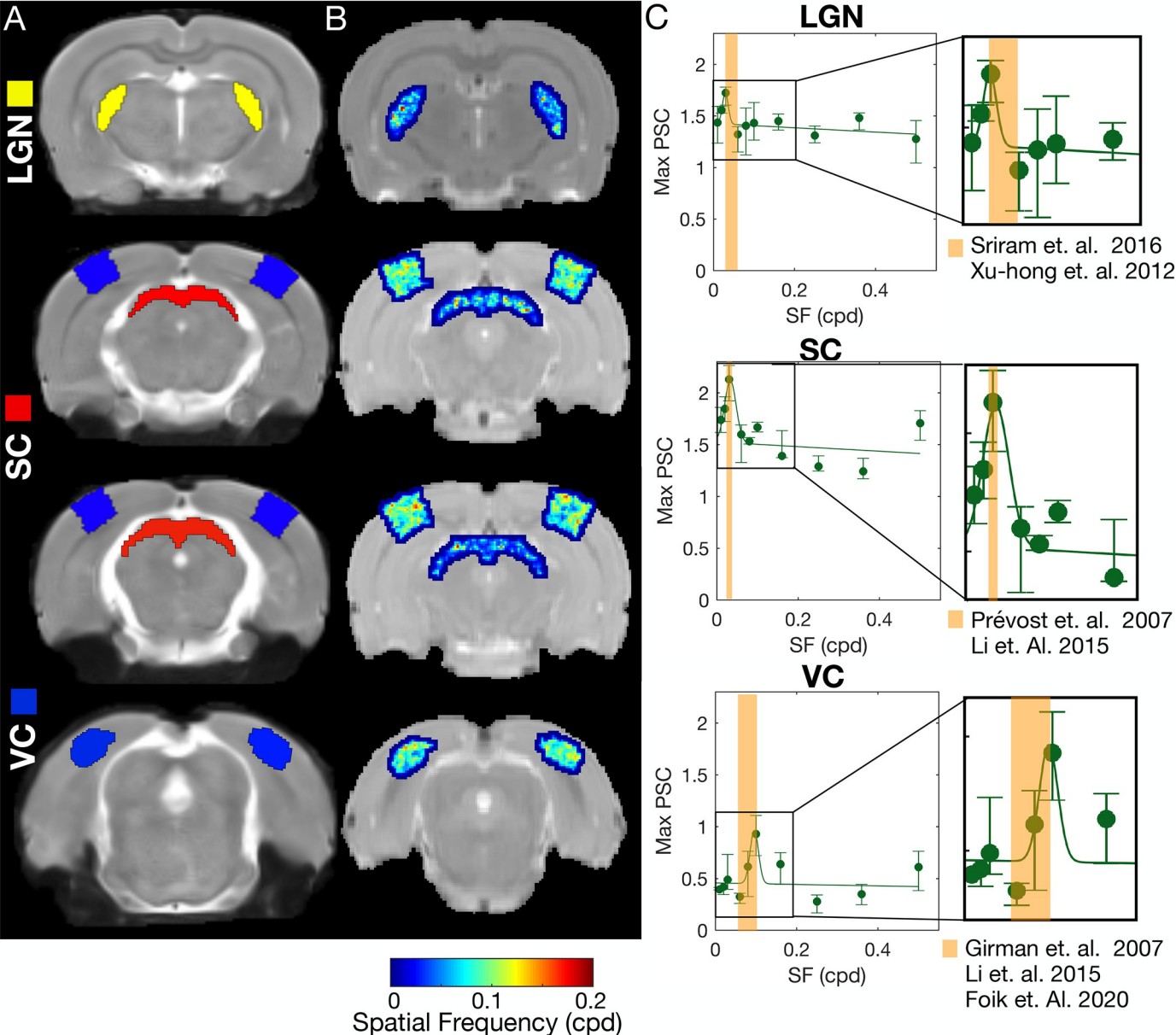

**Fig 3. Spatial frequency selectivity across the visual pathway in HC animals at t = 0.** (A) Anatomical images with the ROIs highlighted. (B) Optimal SF estimated per voxel for HC, averaged across animals. (C) Maximum PSC during the activation period as a function of the SF of the stimulus, calculated for HC. The error bar represents the 10% confidence interval across animals. The continuous lines represent the Gaussian model fitted to the data. The goodness of fit is shown in Table D in S1 Text. The orange band denotes the range of optimal SF values reported in the literature measured using electrophysiology. A compilation of 22 studies reporting on the optimal SF of the rat and mouse visual pathway can be found in Table C in S1 Text. The data underlying this figure can be found here: doi:10.18112/openneuro. ds004509.v1.0.0. HC, healthy control; PSC, percentage signal change; ROI, region of interest; SF, spatial frequency.

according to the SIGMA brain atlas overlaid on top of the anatomical images of a particular HC animal. The projection of the optimal SF per voxel is shown in Fig 3B. In contrast with the retinotopic maps, within each visual structure (i.e., within the ROIs chosen), we find no organization of spatial frequency selectivity. However, the fine grained SF tuning selectivity organization might be masked by coarse resolution of fMRI acquisitions. Fig 3C shows the SF tuning curves in LGN, SC, and VC in HC. Overall, all the ROIs show a band-pass filter tuned to low SF behavior. Early areas of visual processing such as SC have lower optimal spatial frequency than areas that process visual information at a later stage in the visual hierarchy, such as VC. The variation in spatial frequency across visual areas is most likely associated with the different filtering behaviors that diverse neurons exhibit, i.e., some neurons present a low-pass filter behavior, while others exhibit a band-pass [41]. The average optimal SF estimated for VC, LGN, and SC of HC animals is 0.1 cycles per degree (cpd), 0.03 cpd, and 0.03 cpd, respectively. The optimal SF values are in agreement with what has been reported in the literature through calcium and electrophysiology: neurons in the LGN region of awake rats best respond to spatial frequencies of 0.03 to 0.06 cpd [41,42]; neurons in VC have a peak response at 0.1 cpd [28,43,44]; and neurons in SC show band-pass profiles with an optimal spatial frequency of 0.03 cpd and large tuning widths [45]. These reference values are highlighted in Fig 3C.

## 2.3. Visual deprivation results in differential BOLD dynamics throughout the visual pathway

Once we verified that the complex stimuli setup provides insight into the visual pathway organization, we sought to probe the plasticity/stability balance in the adult brain. We first evaluated global fMRI responses in the VC, LGN, and SC of rats that underwent visual deprivation (VD) as a model of plasticity and compared them to activity in healthy controls (HC) in terms of retinotopic and spatial frequency characteristics at multiple time points (t = 0, t = 7d, t = 17d, and t = 27d, Fig 4 and Fig C in S1 Text, respectively).

Upon first exposure to light in animals VD from birth, the fMRI responses to the retinotopy stimulus at this t = 0 were characterized by a markedly faster onset in the VD group when compared with HC in VC and LGN. In the first 10 s of visual stimulation (the same period of stimulation as in the spatial frequency tuning experiment), the BOLD response in VC in the VD group was significantly higher than in HC, t(10) = −5.47, $p$-val = 0.01 (Fig 4C and 4D). In the VC in particular, stronger differences between the VD and HC were observed in response to the spatial frequency tuning stimulus (Fig C in S1 Text). The VD group BOLD responses to the SF tuning stimulus exhibited 3-fold increases in BOLD amplitudes in VC compared to HC at t = 0 ($p$-value <0.001, Fig C in S1 Text). Interestingly, 1 week after light exposure, the VD BOLD responses in the VC were attenuated to the level of the HC for both retinotopic and spatial frequency tuning stimuli. At t = 7d, t = 17d, and t = 27d, there were no significant differences between the amplitude of the VC BOLD responses of VD and HC animals (Fig 4E–4M and Fig C in S1 Text).

Moreover, the HCs show stronger signals in the LGN during the entire stimulation period in response to the retinotopic stimulus at t = 17d, and t = 27d (Fig 4D, 4J and 4M, respectively). Interestingly, the most striking difference between the responses to both stimuli takes place in the SC. For all time points, immediately after the initial overshoot the VD SC exhibited a negative BOLD response to the retinotopic stimulus, contrasting with the positive BOLD response measured in HC, as it is evident in the BOLD activation maps and in the BOLD time series. Note that BOLD signal in SC (key player in saliency detection) has 3 distinct phases, an initial positive peak (likely representing the detection of stimulation onset), a positive plateau phase during the repetitive stimulus duration, and another post-stimulus peak (likely representing a

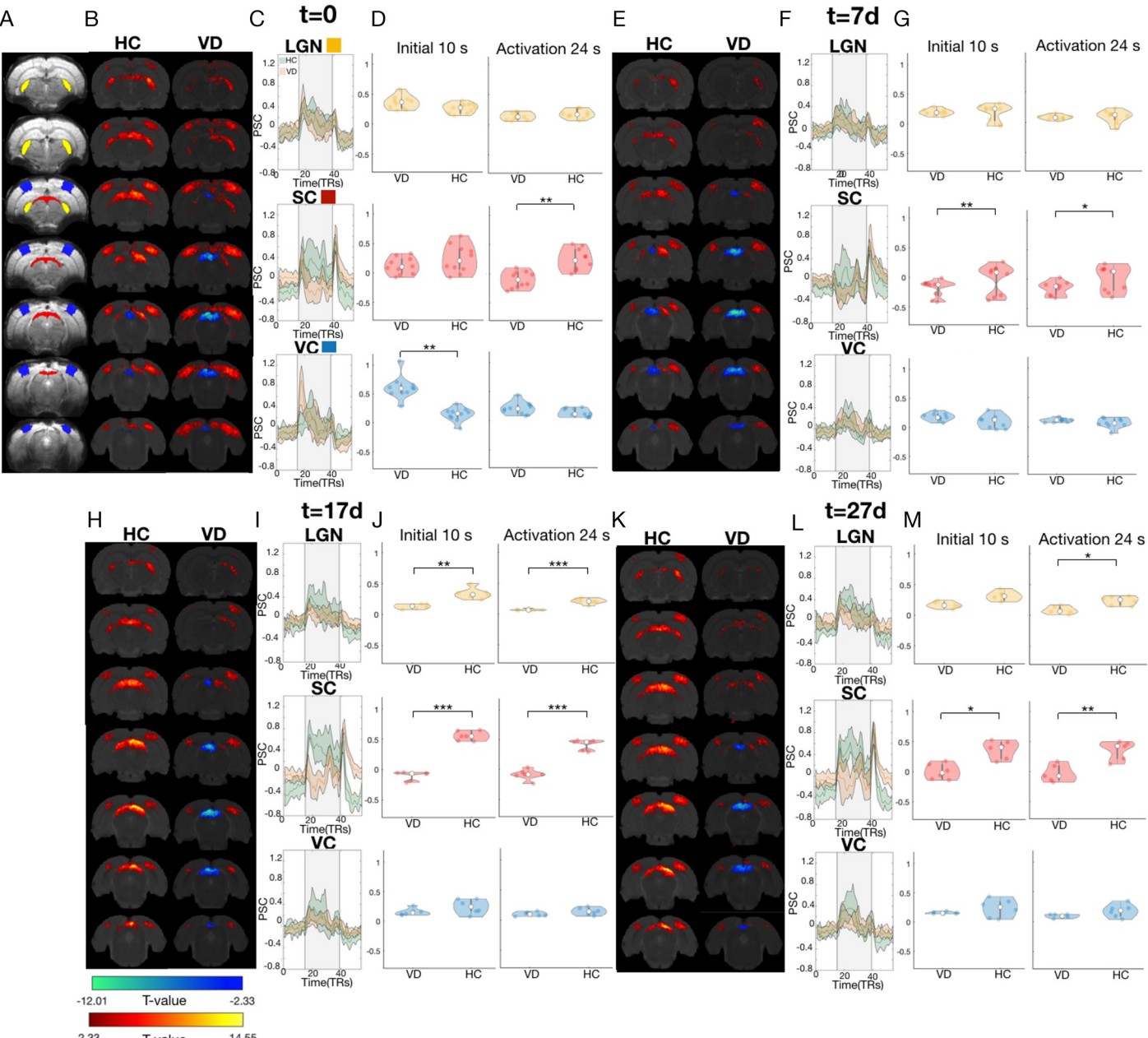

**Fig 4. Differential responses between VD animals and HC driven by the retinotopic stimulus.** (A) Raw fMRI images with the ROIs (LGN, SC, and VC) overlaid. (B, E, H, K) fMRI activation patterns of t-contrast maps obtained for HC and VD animals at t = 0, t = 7d, t = 17d, and t = 27d, respectively. The GLM maps are FDR corrected using a *p*-value of 0.001 and minimum cluster size of 20 voxels. (C, F, I, L) PSC of the LGN, SC, and VC for the HC (green) and VD (orange) animals at t = 0, t = 7d, t = 17d, and t = 27d, respectively. The gray area represents the stimulation period. (D, G, J, M) Violin plot of the amplitude of the BOLD response of VD and HC during the initial 10 s of the activation period (left) and the total duration of the activation period obtained with the retinotopy stimulus (right) at t = 0, t = 7d, t = 17d, and t = 27d, respectively. The white dot represents the mean, and the gray bar represents the 25% and 75% percentiles. The yellow, red, and blue colors represent the LGN, SC, and VC, respectively. The *** represents a *p*-value <0.001, ** *p*-value <0.01, and * *p*-value <0.05. The *p*-values are reported in Table A in S1 Text. The data underlying this figure can be found here: doi:10.18112/openneuro.ds004509.v1.0.0. fMRI, functional magnetic resonance imaging; HC, healthy control; LGN, lateral geniculate nucleus; PSC, percentage signal change; ROI, region of interest; SC, superior colliculus; VC, visual cortex; VD, visual deprivation.

signaling of cessation of activity) [46]. In Fig 4D, 4G, 4J and 4M, we averaged the BOLD signal across the entire stimulation period, merging the initial phases of the SC signal (initial overshoot and plateau). Even when averaged across the entire stimulation block, which includes

the BOLD overshoot after the stimulus onset, this difference was highly statistically significant in all time points (Fig 4D, 4G, 4J and 4M). This contrasts with the responses obtained in SC to the spatial frequency tuning stimulus, where the VD BOLD response is positive at all time points and even shows increased values compared to the HC at t = 27d (Fig C in S1 Text).

To summarize, visual deprivation: (1) boosts BOLD-fMRI responses and results in faster onset times in the visual cortex; and (2) results in negative BOLD responses in the SC in response to the retinotopic stimulus. Moreover, following light exposure the responses in the VC of VD animals are comparable to the ones of HC and remain attenuated in the SC and LGN.

## 2.4. Large-scale and pathway-wide topographical remapping in adulthood following visual deprivation

To gain more specific insights into the reorganization of the adult visual pathway in the brain, we longitudinally tracked changes in pRFs position and size for HC and VD. Fig 5A shows the average phase maps across HC and VD animals obtained for VC, LGN, and SC, respectively. At t = 0, it is evident that while the HC group displays clear retinotopic organization, the VD topography is highly disorganized (the phase maps are scrambled). Interestingly, after first light exposure in the adult VD group, the visual pathways progressively become retinotopically organized and start to resemble the retinotopic organization observed in HC. This progressive organization is particularly apparent in SC (Fig 5). To quantify the topographical remapping, the variation of pRF polar angle as function of cortical thickness across the phase gradient axis, defined as shown in Fig F in S1 Text, was calculated for HC and VD animals for all the scanning sessions. Fig 5B shows that while HC animals show that phase varies with cortical distance consistently across all scanning sessions, for VD animals at t = 0, the relation between pRF and cortical distance is nearly random, which is evidenced by the flat slope. With light exposure, the relation between VD animals pRF phase and cortical distance becomes more apparent and the slope progressively increases in magnitude. Fig 5C shows that at t = 0, the magnitude of the slope between phase variation and cortical distance is significantly higher for HCs than for VDs for VC, LGN, and SC. Whereas at t = 27d HCs and VDs show nearly identical slopes for all visual areas.

Next, we investigated whether the progressive organization of the visual pathways is accompanied by a decrease in pRF size. Fig 6A shows the average pRF size across animals and ROIs for HC and VD at the different time points. At t = 0, the pRF size in VC and SC is significantly larger in VD than in HC. One week after light exposure, the VD's pRFs in VC, LGN, and SC are significantly smaller compared to its values at t = 0. For VC and SC at t = 7d, t = 17d, and t = 27d, the VD's pRF values were at the level of HC. In LGN, VD's pRF size gradually shrinks across scanning sessions. In addition the pRF size in HC was larger than in VD at all the time points measured, this effect becomes significant at t = 17d and t = 27d. Importantly, the HC pRF estimates for VC and SC did not significantly differ between scanning sessions. In LGN, however, from t = 0 to t = 7d, there is a significant reduction in pRF size for HC. Furthermore, the averaged pRF size estimated per visual area in HC (VC = 21˚; LGN = 32˚; SC = 41˚) are in line with values reported in the literature (note that our model assumes simple on-center RFs) [47–49]. This provides further evidence of reliability of the pRF estimates. Fig 6B shows the variation in pRF size across VC cortical layers for HC and VD at multiple scanning sessions. Although the variation of pRF size as function of cortical depth varies across scanning sessions, in particular in VD, the reduction of pRF sizes took place mostly in the superficial and middle layers. At t = 0, the pRF sizes at the deepest layers do not differ significantly between HC and VD. Furthermore, at t = 0, the variation of the pRF size across cortical layers does not follow

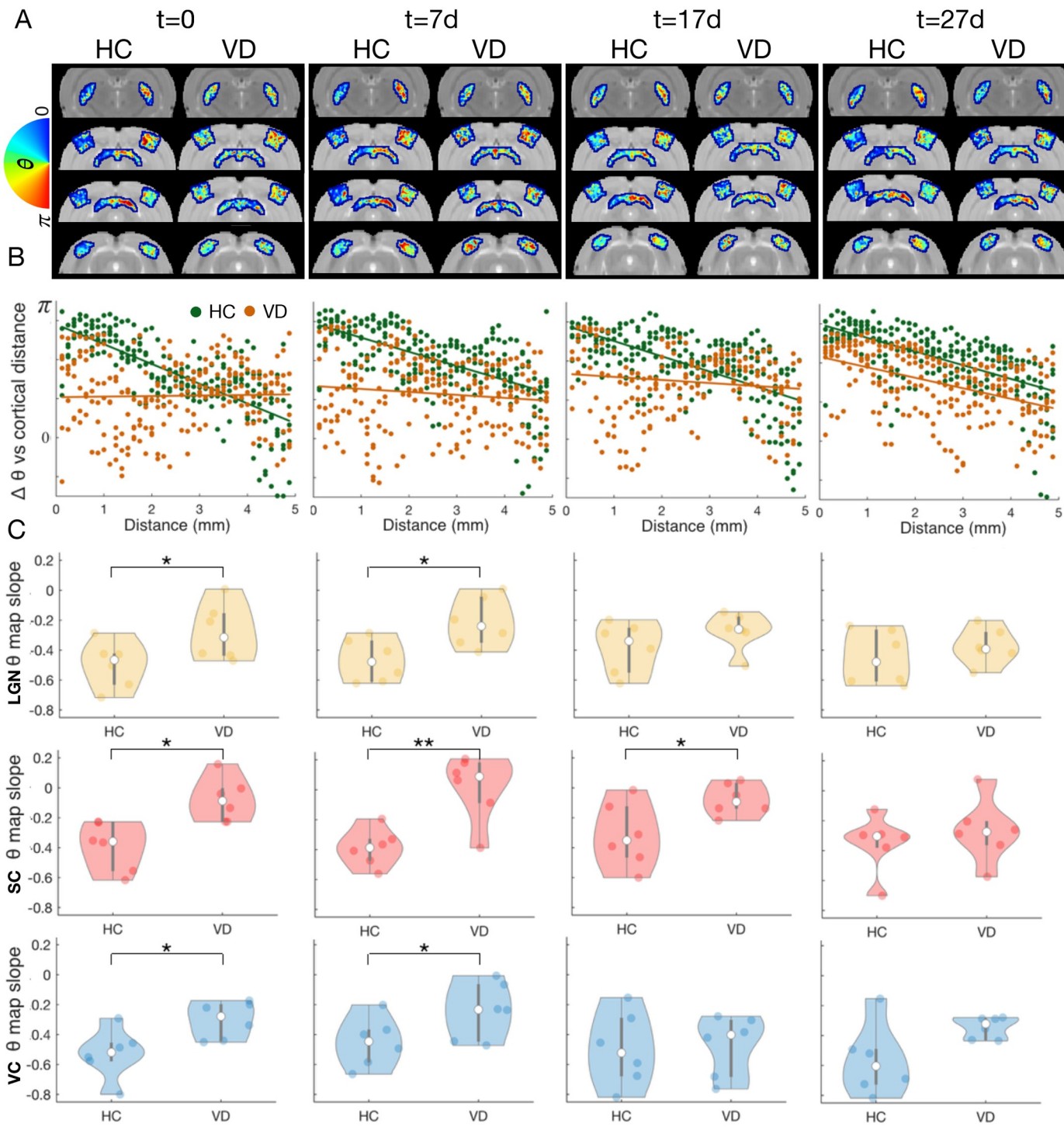

**Fig 5. Refinement of RF position across time for VD and HC.** (A) Average phase maps obtained for 4 different slices of VD and HC at the 4 measured time points (t = 0, t = 7d, t = 17d, and t = 27d). (B) Variation of the pRF estimated phase as function of the cortical distance across gradient direction, shown in Fig F in S1 Text, measured for the SC of HC (green) and VD (orange) across multiple time points (t = 0, t = 7d, t = 17d, and t = 27d). The green and orange lines correspond to the linear fit for HC and VD, respectively. (C) Violin plot of the slope of the correlation between the pRF phase variation and cortical distance measured for HC and VD, for each ROI (LGN, SC, and VC) for multiple time points (t = 0, t = 7d, t = 17d, and t = 27d). Each dot corresponds to a different animal. Only the animals that performed the 4 scanning sessions were included in the analysis. The *** represents a *p*-value <0.001, ** *p*-value <0.01, and * *p*-value <0.05. The data underlying this figure can be found here: doi:10.18112/openneuro.ds004509.v1.0.0. HC, healthy control; LGN, lateral geniculate nucleus; pRF, population receptive field; RF, receptive field; ROI, region of interest; SC, superior colliculus; VC, visual cortex; VD, visual deprivation.

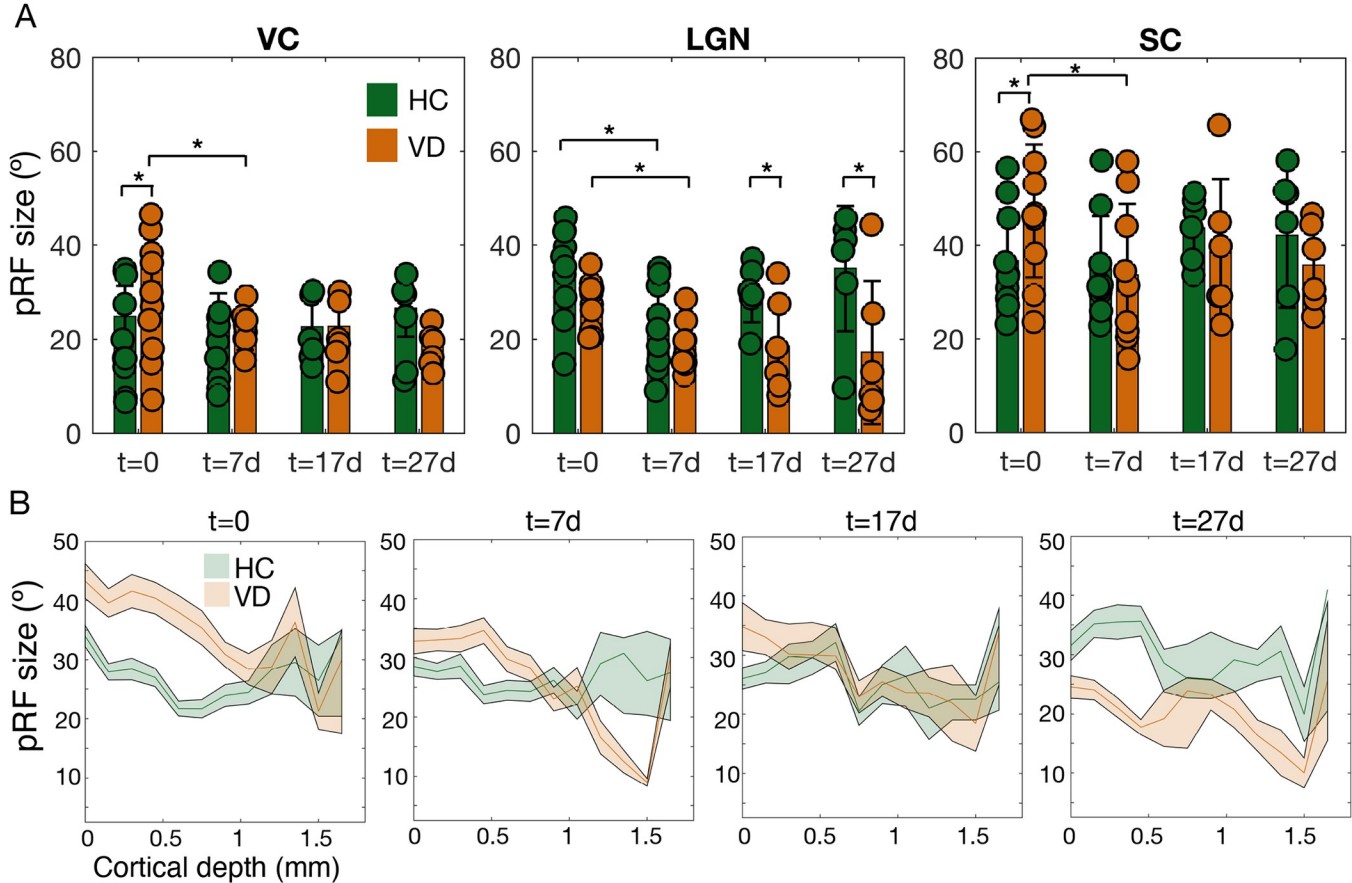

**Fig 6. Refinement of RF size across time for VD and HC.** (A) Average pRF size measured for HC and VD at multiple time points for VC, LGN, and SC. The error bar corresponds to std. The ** represents a *p*-value <0.01 and * *p*-value <0.05. The statistical analysis was performed using ANOVA Bonferroni corrected for multiple comparisons (brain area and scanning session). The *p*-values are detailed in Table B in S1 Text. (B) Variation of the pRF size averaged across animals as a function of the cortical depth for 2 slices of the VC for VD and HC at multiple time points. The data underlying this figure can be found here: doi:10.18112/openneuro.ds004509.v1.0.0. HC, healthy control; LGN, lateral geniculate nucleus; pRF, population receptive field; RF, receptive field; SC, superior colliculus; VC, visual cortex; VD, visual deprivation.

the trend of the HC; in the VD, the pRF size continuously decreases from the cortical surface until the deepest layer.

## 2.5. Specialization of spatial frequency selectivity is promoted by visual experience

A hallmark sign of specialization of the visual pathway is the refinement of tuning curves [50]. Fig 7A shows the projection of the optimal SF obtained per voxel. At t = 0, a wider range of spatial frequencies are apparent for the VD group VC, LGN, and SC when compared to the same structures in HCs, while at t = 7d, the VD maps are more similar to HC's. Fig 7B represents the spatial frequency tuning curves in VC, LGN, and SC at the 4 time points tested. Panels 7C and 7D show how the optimal spatial frequency (Gaussian center) and broadness of the tuning curves (Gaussian width) are shaped by visual experience in HC and VD. For all the visual structures tested (VC, LGN, and SC) at t = 0, the VD tuning curves are broader than HC ones (VC t(39) = −2.48 *p*-value <0.01, LGN t(39) = −3.7 *p*-value <0.001; SC t(39) = −2.61 *p*-value <0.01); panel D. At t = 7d, the VD spatial frequency tuning curves become similar to HCs, narrower and with an aligned peak (optimal spatial frequency). Note the drop in optimal

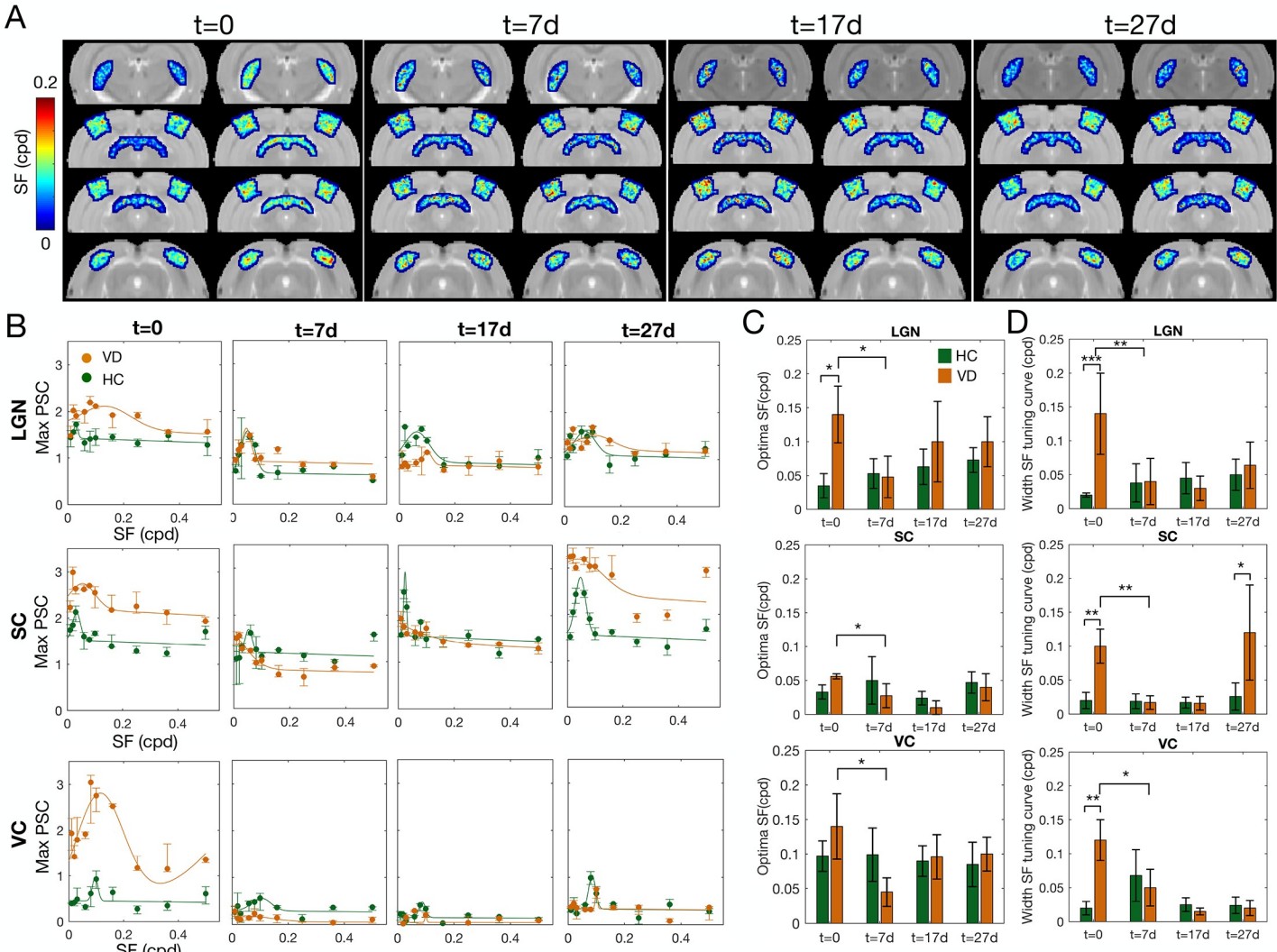

**Fig 7. Visual experience following VDM promotes the specialization of spatial frequency tuning curves.** (A) Optimal SF estimated per voxel obtained for 4 different slices of HC and VD, at t = 0, t = 7d, t = 17d, and t = 27d, respectively. (B) Maximum PSC during the activation period as a function of the SF of the stimulus, calculated for each ROI (LGN, SC, and VC) of VD (orange) and HC (green) at the 4 measured time points. The error bar represents the std. The continuous lines represent the Gaussian model fitted to the data. (C, D) Estimated optimal SF (Gaussian center, panel C) and broadness of the SF tuning curve (Gaussian width, panel D) for the LGN, SC, and VC at t = 0, t = 7d, t = 17d, and t = 27d for HC and VD. The error bar corresponds to std. The *** represents a *p*-value <0.001, ** *p*-value <0.01, and * *p*-value <0.05. The data underlying this figure can be found here: doi:10.18112/openneuro.ds004509.v1.0.0. HC, healthy control; LGN, lateral geniculate nucleus; PSC, percentage signal change; ROI, region of interest; SC, superior colliculus; SF, spatial frequency; VC, visual cortex; VD, visual deprivation; VDM, visual deprivation model.

spatial frequency and spatial frequency tuning curve width from t = 0 to t = 7d in panels 7C and 7D. This is particularly evident in the LGN that shows a much higher correlation between the VD and HC tuning curve points at t = 7d than at t = 0, (t = 0 r² = 0.11, t = 7d r² = 0.73). While at t = 17d and t = 27d, the values of optimal spatial frequency estimated for VD tend to stabilize near the spatial frequency values estimated for HC, the broadness of the tuning curve is more variable. After the initial shrinkage in spatial frequency tuning broadness from t = 0 to t = 17d, at t = 27d SC exhibits larger tuning curves than HC, panels 7B-D. With the exception of LGN where the HC's spatial frequency curves become larger over time and the optimal spatial frequency is shifted toward higher values, the optimal spatial frequency and broadness of

the tuning curve estimated for VC and SC for HC do not vary significantly between time points. In addition, in areas with strong BOLD signals such as SC, the spatial frequency estimates tend to be less variable.

Note that in all the sessions, the baseline remains high, in particular, for the LGN and SC. This is likely due to the fact that there are responsive voxels to all the spatial frequencies tested and neurons in VC, LGN, and SC show a high variability in the spatial frequency tuning properties. In particular, the spatial frequency cut off (a measure of the spatial resolution) in the rat LGN, SC, and VC varies between the following intervals ([0.01 and 1 cpd]; [0.02 1.2 cpd]; [< 1.2 cpd]) [28,41,45]. This covers the entire range of spatial frequencies that we used to estimate the tuning curves. Therefore, the fMRI-based spatial frequency tuning curves shown in this study correspond to the SF interval to which there is stronger overlap between the voxels response (spatial frequency range that most voxels respond strongly).

## 3. Discussion

The unique setup for delivery of complex stimuli in rodent preclinical fMRI scanners developed here enabled an in-depth investigation of the plasticity/stability balance in the adult visual pathway. Delivery of such complex stimuli enabled the first fMRI-based mapping of retinotopic organization and SF tuning of the visual pathway in adult rats. In HCs, the estimation of the RF properties via BOLD-fMRI reproduced at a pathway-level all the trends expected from prior invasive calcium recording and electrophysiology studies [27,28,51] including: retinotopic organization, RF size, variation of VC RF sizes with cortical depth, and SF tuning curves. Hence, BOLD-fMRI signals faithfully represent these specific features while offering a very high resolution, longitudinal investigation, and a comprehensive whole-pathway vantage point. To the best of our knowledge, our work is the first fMRI-based retinotopic mapping and SF tuning curves in the rat visual pathway, and we note that the high-resolution ($125 \times 125$ $\mu$m$^2$) population RFs that could be mapped using this approach can probably be extended in the future using denoising techniques [52] and/or adaptation of the system to cryogenic coils [53]. This represents a leap forward compared with advanced techniques such as high density recordings (e.g., with chronically implanted neuropixels) or longitudinal calcium imaging. First, fMRI is a noninvasive technique; the introduction of probes in the brain can lead to bias in the estimation of RFs that are erroneously interpreted as plastic changes. In addition, the cell swelling resulting from inflammation post-introducing the probe can mask plastic changes. Second, fMRI (and in particular, ultra-high field MRI with the use of cryoprobe) allows to measure whole brain neural activity simultaneously with good SNR from the cortex to deeper structures, this is not possible with any other technique. Electrophysiology is limited to particular brain areas and wide-field calcium imaging is limited by the cranial window, curvature of the cortex, and the scattering of light in the brain (which results that superficial layers of the cortex have a higher contribution to the wide-field calcium signals than deeper structures) [54,55].

Once all the important features of the visual pathway were reproduced using BOLD-fMRI contrast, we could harness this approach towards investigating plasticity and stability in the visual pathway—in particular, how VDM affects the function and organization of the brain and how the visual experience promotes reorganization. Our main findings are: (1) the adult rat brain is highly plastic and visual deprivation delays the maturation of RFs, in particular, visually deprived animals lack retinotopic organization and spatial frequency specialization of the visual pathway; and (2) light exposure during adulthood (post-critical period) clearly promoted an extensive topographic remapping and functional specialization of the visual system in VD rats. At the level of BOLD signals, we show that the VDM affects the activation

magnitude and timing of the responses. Then, using biologically inspired computational models of the RFs applied to the BOLD signals in response to the complex stimuli, more perceptual features of the pathway could be unraveled, including that exposure to light upon visual deprivation opens a window of plasticity during adulthood that promotes specialization of the pathway towards normal vision. This unique view of the whole pathway is promising for future characterizations of plasticity in health and disease. In a clinical context, this technique holds the potential to assess the optimal timing for visual restoration and rehabilitation therapies, such as retinal stem cell transplantation, and to be pivotal for the translation of preclinical findings to humans. Below, we discuss our findings and their implications in detail.

## 3.1. Visually deprived rats exhibit different BOLD response patterns compared to healthy controls which tend to normalize following visual experience

**3.1.1. VC and LGN: Visual deprivation model modifies BOLD-fMRI response's amplitude and timing.** The VC and LGN of visually deprived rats from birth, at the first moment of light exposure (t = 0) showed an early onset time and stronger BOLD responses compared to normal reared rats in the VC. This may reflect: (1) changes in excitation/inhibition balance of the visual cortex, most likely a reduction of inhibitory signals [56,57]; and (2) an adaptation mechanism through increase of contrast-gain, which results in enhanced excitability of the visual cortex following visual deprivation [58]. This cortical increase in gain-control reflects the adaptation mechanisms attempting to optimize weak or absent information during the visual deprivation period. Furthermore, similar mechanisms have also been described in humans, in which short-term monocular patching boosts the patched eye's response in the visual cortex [59,60] and short-term binocular visual deprivation increases the excitability of the visual cortex [58]. However, our results are in contrast with a previous study using ultrafast fMRI that showed delayed, broad, and low amplitude BOLD responses measured in the VD mice when compared to HCs [33]. There are multiple factors that contributed to the different results between these two studies: (1) differences between the stimulus. There are major differences between the visual stimulus presented to the animals. Gil and colleagues (2021) used simple low frequency flickering visual stimuli and the visual stimulation was monocular [33]. Here, we used far more complex stimuli: the duration of stimulation, stimulus size and content of the stimulus, the contrast, the spatial frequency, and the movement direction very likely influenced the dynamics of the measured BOLD-fMRI responses and are likely to originate the differences between the studies. Gil and colleagues (2022) varied the temporal flickering frequency of the LED visual stimulation, this resulted in dramatic changes in the BOLD response profile across the entire visual pathway (including the changes in the positiveness of the BOLD responses) [46]. The fact that the change of a single stimulus parameter can result in such dramatic changes in the BOLD responses suggests that the different findings between this study and the one of Gil and colleagues (2021) can be attributed to the different types of stimulus used. We believe that more complex and well-defined stimulus with longer duration will elicit more reliable BOLD responses, and they are better suited to probe plasticity mechanisms; (2) experimental design (in this study, we used multislice acquisitions (12 slices) at TR 1.5 s versus ultrafast (50 ms) single slice acquisition Gil and colleagues (2021)); and (3) species used (rat versus mice, which present significant differences in the processing of visual information (Fig E in S1 Text) and respond to anesthesia differently).

Furthermore, we examined the BOLD-fMRI responses at multiple time points following light exposure. After 1 week of light exposure, the BOLD responses to the retinotopy and SF tuning stimuli in the VD group returned to the level of HCs, reflecting that the window of

plasticity (or critical period) in VD rats has been extended. This extension could be mediated by modification of excitatory–inhibitory balances [61] and/or removal of brakes on plasticity [62].

**3.1.2. SC: Visual deprivation elicits negative BOLD responses to retinotopic stimuli in superior colliculus.** For all the time points, the SC in dark reared animals showed a different behavior to that described above for VC and LGN. In particular, SC exhibited a negative BOLD response contrasting with the positive BOLD response measured in HC. Although interpretation of negative BOLD responses is ambiguous, mounting evidence suggests that negative BOLD responses correlate with local decreases in neural activity [63]. One possible explanation for the negative BOLD responses observed in the VD group is enhanced intracortical or tecto-tectal inhibition [64]. Previous studies have shown that visual deprivation potentiates inhibitory feedback and that it reflects a degradation of the visual function [65,66]. The fact that the enhanced intracortical inhibition mostly affects SC is likely linked with the fact that SC receives mostly direct retinal inputs and it has a classical RF with excitatory center and an inhibitory surround [45,67]. The inhibitory inputs lead to an enhancement of surround suppression. Some studies suggest that early but not late light exposure protects against the effects of adult visual deprivation on the SC [68]. Furthermore, in a recent study Gil and colleagues investigated the continuity illusion phenomenon in the rat, through behavior, fMRI, and electrophysiological recordings. They found that when continuity illusion is achieved, the SC fMRI signal changes from positive to negative regimes. Importantly, electrophysiological recordings in SC point to neuronal suppression as a source of the negative BOLD signal [46]. This study also points to the idea that chronic visual deprivation disrupts the activation–suppression balance in SC, leading to negative BOLD responses in the SC in VD animals throughout the 4 scanning sessions.

**3.1.3. Differential vascular responses are not the underlying sources of the differences in BOLD responses observed between VD and HC.** Negative BOLD can also be driven by reduced cerebral blood volume, a mechanism known as "vascular stealing" [69]. To discard that differences in vasculature between VD and HC are the dominant driving force for the differential BOLD-fMRI responses observed between the 2 groups, a hypercapnia challenge was performed. Hypercapnia is a strong vasodilator, causing increases in cerebral blood flow and cerebral blood volume and it is used to calibrate BOLD fMRI [70]. We found that the rise times and signal amplitude were nearly identical between VD and HC for the different ROIs, (Fig I in S1 Text). This suggests that vascular responses are similar between the groups and further pointing towards the differences being driven mainly by neural activity.

## 3.2. Refinement of receptive fields following light exposure during adulthood in visually deprived animals across the entire visual pathway

**3.2.1. Mapping noninvasive high-resolution population receptive field properties.** The VC does not receive a direct input from the retina. The activity of rodent VC neurons is driven by the geniculate pathway that projects feedforward visual information to layer IV of VC and by SC (which receives direct input from the retina) that projects broadly to all cortical layers of VC [71]. Therefore, the ability of cortical neurons to integrate visual information varies across visual areas [45,67]. This notion is in line with our results, as we found that the superficial layer of SC has the largest RFs (approximately 41˚), while VC and LGN have smaller sizes, 21˚ and 32˚, respectively. The pRF sizes estimated are in line with electrophysiology studies, which estimated average VC RF sizes of approximately 20˚, and SC RFs ranging from 10˚ to 60˚ [28,67,72]. Regarding the RF profiles across cortical layers, in the VC the population RF sizes are the largest at most superficial and deepest layers and smallest at layers IV and V. This is in agreement with electrophysiological studies in cats and human high-field fMRI [39,40]. The

variation of the RF across cortical layers is linked to the flow of signals across the cortical architecture [39].

**3.2.2. Visual experience following dark rearing promotes a pathway-wide receptive field remapping and specialization (shrinkage).**   We mapped the specialization of the visual pathway in VD rats during light exposure. Immediately after light exposure, VC, LGN, and SC lacked retinotopic organization, the phase maps were totally disorganized when compared to the HC ones. With the continuous light exposure, the visual pathway progressively became retinotopically organized, reaching the same level of retinotopic organization as HC 1 month after light exposure. Across the entire visual pathway, the progressive organization of the visual pathways was accompanied by a shrinkage in pRF sizes. The pRF shrinkage took place during the first week of light exposure, and it stabilized between t = 7d and t = 27d. The fact that VD's RF size stabilizes before the retinotopic structure suggests that the shrinkage of RF is the driving force for the reorganization of the retinotopic structure. The decrease in RF size is consistent with the time course of visual acuity development. Our results agree with previous electrophysiology and calcium studies that locally showed that, in VD adult animals, the RF sizes of visual cortical neurons remain large [4,15,73]. Furthermore, the refinement of RFs and SF curves corroborates the notion that dark rearing slows down the maturation of brakes on plasticity and that visual experience acts by modulating the level and the patterning of neural activity within the visual pathways.

## 3.3. Visual experience refines spatial frequency tuning curves

To investigate how visual deprivation affects the functioning of the visual pathways, we estimated for the first time the SF tuning curve across the rat visual pathway using fMRI. In general, the neural populations in VC, LGN, and SC respond to spatial frequencies between 0.02 and 0.16 cpd, with an average optimal spatial frequency estimated for VC, LGN, and SC of 0.1, 0.03, and 0.03 cpd, respectively. The optimal SF estimates are in line with electrophysiology studies, Table C in S1 Text, and they link with the RF size estimated. It is assumed that there is an inverse relation between RF size and RF spatial frequency sensitivity, so that neurons with large RF sizes process mainly coarse spatial information. Here, we show that VC, which processes higher spatial frequencies, has smaller RF sizes than LGN and SC that process lower spatial frequencies. SC and LGN are subcortical relays of visual information, their function is not only to receive the direct projections from the rat retina and pass the information to the visual cortex where high order processing takes place, but also they are involved in multiple functions related to fast responses to the visual stimulus, such as orienting the body and eye movements towards the stimulus and guiding spatial movement using visual information [74–76]. Therefore, they benefit from large RFs tuned to low SFs, which provide coarse information about the visual environment and facilitate motion detection. In addition, lesion studies have shown that lesioning the SC does not impair the spatial frequency perception nor spatial acuity [77]. This suggests that high spatial frequency is predominantly processed in the cortical visual areas.

Furthermore, we tracked the dynamics in the SF tuning curves in HC and VD from the first moment to 27 days after light exposure. We found that when first exposed to light, the VD rats show broader tuning curves than HC. One week after light exposure the tuning curves became sharper (narrower and with an aligned peak with HC tuning curves), which suggests an increase in selectivity. This was evident in SC, LGN, and VC. Note that the tuning curves obtained for HC are stable over time. These findings are in line with calcium and electrophysiology studies in mice which have shown: (1) that spatial frequency shifts are accompanied by a decreased tuning bandwidth [78]; and (2) the VD delays the maturation on the cutoff spatial frequency (highest sinusoidal grating frequency detectable by the visual system) [43,79].

However, Zhang and colleagues showed that VD mice showed lower spatial frequency cutoffs compared to HCs, whereas our results indicate that at t = 0 VDs have higher spatial frequency cutoffs than HCs. Furthermore, the refinement of SF tuning curves tightly links with the results regarding the refinement of the RF over time, once large RF have a preference for low SFs and vice versa. Similar patterns were also observed in electrophysiology studies where the topographic refinement was accompanied by a decrease in spot size preference and an increase in surround suppression [5].

The lack of/diminished activity in the VC at t = 17d and t = 27d for HC and VD animals in response to the SF stimulus likely reflects adaptation to this particular stimulus rather than a bias arising from the experimental setup, acquisition, visual stimulus, or analysis. For the following reasons: First, during the same sessions (t = 17d and t = 27d), the animals exhibited clear activation in the visual cortex in response to the retinotopic stimulus (Fig 4). Apart from the VD animals that exhibit a reduction in BOLD activation in the VC from t = 0 to t = 7d, the BOLD signal in the VC did not vary between sessions, see Table A in S1 Text. This shows that during the same scanning session, the VC is responsive to retinotopic stimulus and remains consistent across all scanning sessions. Given that we performed all the experiments in the same session using the same setup for the display of the spatial frequency stimulus and retinotopic stimulus, it is unlikely that the lack of BOLD response to SF stimulation represents experimental/technical variabilities across time. Second, the animals were induced using isoflurane, which is a vasoactive drug that affects the basal BOLD signal [80]. As the spatial frequency stimulus was presented after the retinotopic stimulus, the remaining isoflurane effects should have been washed away during scan preparation and anatomical image acquisitions, and therefore, are highly unlikely to be the responsible for the lack of BOLD signal in SF stimulation (while in the retinotopic stimulation, as mentioned above, the BOLD responses were preserved). Third, the stimuli characteristics did not change between sessions. Therefore, this cannot explain variability between sessions. Fourth, the fact that LGN and SC show clear activation in the same experiment in which the VC does not exhibit strong activation also proves that the visual stimulation was properly delivered and that the animal physiology was stable. If LGN and SC were only responding to luminance changes instead of spatial frequency, we should expect that the strength of the BOLD response would be constant across spatial frequencies; however, we see that some spatial frequencies elicit a stronger BOLD response. Thus, the setup was correctly working and the animal physiologically stable. Fifth, although the signal is highly reduced in the VC at t = 17d and therefore noisy, it still peaks at the spatial frequency values that correspond to the optimal SF selectivity for VC, again indicating that the few neurons that are active maintain spatial frequency selectivity. Taken together, these points strongly suggest that plasticity or adaptation to the stimulus underlies the lower BOLD modulation in VC in these time points. This effect needs to be investigated in future studies and was clearly beyond the scope of this one, but it highlights the power of using fMRI for investigating the whole pathway using our novel setup.

In summary, our findings suggest that the VDM prevents the maturation of the RFs, as they remain large without spatial frequency selectivity and, simultaneously, that VDM extends the critical period. Visual experience during adulthood promotes the specialization of the RFs to maturation level similar to the one of healthy controls.

Our study points towards long-term and slow (in the scale of hours to days) RF modulations that span the entire visual pathway. This large-scale remapping is likely regulated by a combination of Hebbian (more prominent during the deprivation period) and homeostatic plasticity (overruling upon visual experience) that allows the neurons to return to its specialized state following VDM. This is in line with the view that immediately after birth, brain specialization and development is driven by spontaneous activity; however, chronic visual

deprivation progressively leads to less specialized visual neuron responses, which can be reacquired/enhanced by light exposure (even in adulthood), as we have shown in this study. Furthermore, this theory is corroborated by studies on synaptic scaling, changes to the balance between excitation and inhibition [81] and by theoretical frameworks that show that homeostatic mechanisms regulate the interplay between excitation and inhibition and may be the driving factors for long-lasting macroscopic changes and alterations of functional networks [82].

### 3.4. Limitations and future research

During scanning, the animals were sedated. Although medetomidine has been shown to be suitable for longitudinal studies [83], it can introduce bias compared to the awake state. One way to eliminate the anesthesia effect is to perform the experiments in awake animals; however, this requires prolonged training to scanner noises and to the visual stimulus (to maintain a stable fixation throughout the experiment) which would be very difficult in VD animals. Another limitation of the study is that although the VD animals were kept in a dark room from birth, husbandry was performed under red light. A recent study has shown that although rats do not possess red cones, their visual capacity under red light is still preserved [84]. Hence, the animals are not "completely" deprived of light, but their visual pathways can still be expected to be highly immature. Eye movements during retinotopic mapping contribute to unreliable RF estimates; in particular, larger pRFs may be biased due to unstable fixation [85]. While we did not record eye movements during scanning, the animals were sedated with medetomidine, which is widely accepted as effective in reducing spontaneous eye movements, as well as eye movements in response to visual stimuli [86]. For example, Nair and colleagues have compared effects of common sedative drugs, and they found that ketamine/xylazine (which similarly to medetomidine is an alpha-2 adrenergic agonist [87,88]) reduced eye movement as effectively as 1% isoflurane with the muscle relaxant pancuronium [89]. Besides being a well-established sedative for fMRI, medetomidine has shown to elicit higher electroretinogram responses than ketamine/xylazine [90]. Due to the long duration of our scanning sessions and the need to have sedated animals with minimal eye movement and strong retinal responses/visual function, we chose medetomidine as the sedative. Thus, the mitigating effects of sedation coupled to our clear observation of very reasonable retinotopic maps suggest that eye movement confounders were significantly attenuated. In addition, spontaneous eye movement cannot be expected to be coherent between different repetitions of stimuli. In each scanning session, we performed 6 separate repetitions of the stimuli, and averaged them, thereby likely washing away most of any potential residual eye movement in the mean signal.

Here, we used an fMRI-based approach to calculate the SF tuning curves and although the optimal SF estimates are in line with electrophysiology, 2 photon imaging and intrinsic signal optical imaging the tuning curve width might be underestimated. There are multiple factors that can contribute to the narrow fMRI spatial frequency tuning curves: (1) they were obtained from the signal of the entire ROI. There are responsive voxels to all the spatial frequencies and the width of the tuning curve corresponds to the spatial frequency interval to which there is stronger overlap between the response of more voxels. (2) The coarse SF sampling (only 10 SF were tested, having more points between 0.03 cpd and 0.12 cpd could have improved the fits); (3) 2 runs of the SF stimulus were averaged; and (4) the spatial smoothing fMRI can also affect the SF tuning fit. Furthermore, despite being very high, the spatial resolution of this study's fMRI acquisition (125 × 125 × 650 μm3) likely masked the fine grained spatial frequency tuning selectivity organization, and it might explain why within each visual structure (ROI), we find no specific organization of spatial frequency selectivity. In fact, recent studies have found that the similarity of spatial frequency tuning decreases as a function of cortical distance,

refuting the idea that the rodent visual pathway is salt-and-pepper organized [91]. In addition, it has been suggested that the lack of spatial frequency selectivity reported in other studies [92,93] could be related with the resolution scale of the measurements, in particular, cortical distances between 100 μm and up to 250 μm, which comprise a large number of cell pairs with disparate direction preferences.

One of the limitations of using fMRI is that it lacks single cell specificity, so experimentally, it is not possible to disentangle whether the pRF refinement arises from single cell RF refinement or from a decrease in RF variability across the population of neurons belonging to the same voxel. In our view, the 2 possibilities are tightly related, as the refinement of individual RFs will lead to a decrease in variability of the RFs properties of neurons captured by the same voxel. Indeed, previous single-cell studies have shown that visual RFs and retinotopic maps develop through individual cell refinement processes driven by fine-scale circuit refinement in which imprecise connections are weakened and eliminated and correctly targeted connections are strengthened and maintained, leading to the maturation of surround suppression in visual neurons [94].

The neural basis of the plastic mechanisms here reported, i.e., whether it is Hebbian plasticity (associated with positive feedback) or homeostatic plasticity (linked with negative feedback) can be further investigated using ultra-fast resting state MRI approached together with connectivity models, i.e., connective field [95] and [96].

Despite the usefulness of BOLD-fMRI, its neural underpinnings are still indirect [97]. Our hypercapnia results suggest that the effects are highly unlikely to be of pure vascular origin. However, to better dissect the relationships of the observed BOLD responses and underlying activity, simultaneous electrical recordings, calcium recordings, and/or optical imaging could be fruitfully applied with our setup [98].

To conclude, our work illustrates how high-resolution fMRI, in combination with structured visual stimulation, can bridge the spatiotemporal scales necessary for repeatedly interrogating structural and functional changes underlying plasticity and longitudinally investigate how the stability/plasticity balance is sculpted by visual experience. Besides its relevance to understand the foundation of vision and plasticity, the findings of this study will form the basis for future assessment of the effect of retinal degeneration on visual function and the efficacy of any therapeutic intervention in animal models of retinal degeneration.

## 4. Materials and methods

### 4.1. Experimental design

All the experiments strictly adhered to the ethical and experimental procedures in agreement with Directive 2010/63 of the European Parliament and of the Council and were preapproved by the competent institutional (Champalimaud Animal Welfare Body) and national (Direcção Geral de Alimentação e Veterinária, DGAV) authorities, under the Plastimap Project, License number 0421 000 000 2021.

**4.1.1. Dark rearing.** In this study, $N = 20$ adult Long–Evans rats ($N = 18$ females, 12 to 20 weeks old, mean weight 322 g; range: 233 to 445 g) were used. The animals were randomly assigned into 2 groups: healthy controls (HC, $N = 10$) and dark reared (VD, $N = 10$). The VD animals were born and kept in the dark until 10 to 12 weeks of age with ad libitum access to food and water. Specifically, the animals were housed in a sound protected dark room. The husbandry and animal preparation for the MRI scanners were performed using red light (for which the animals are less sensitive). The red light consisted of a combination of a red light source (wavelength peak at 620 nm, MASTER TL5 HE Colored 14W Red1S) plus a red filter (RC-3 Light Gard, Rose Chocolate window film spec sheet) that filters out residual spectral

lines between 200 nm and 580 nm. Note that the rat visual sensitivity is estimated to peak at 362 nm and 502 nm for UV and M-cones, respectively, and to be insensitive for wavelengths above 620 nm [99–101]. The animals were exposed to the red light only once a week for less than 2 min (within the dark room, the animals were housed in a rack shielded from light. During the husbandry, 1 clean cage was placed in the dark room and the animals were only exposed to the red light during the process of cage transfer. Once the animals were in the clean cage, the cage was immediately returned to the rack protected from light). Transport to the MRI room and anesthesia were done in the dark, and animal prep for scanning was also performed in the dark with the same red light source and filters as described above. Thus, the animals were first exposed to light during the first MRI scan (t = 0). After the first MRI session, the VD animals were housed in a normal environment with a 12 h light/12 h dark cycle. Follow-up scans took place 7, 17, and 27 days after t = 0. The normal reared HC rats were scanned following the same protocol but were born and kept in the normal environment (12 h/12 h light/dark cycle) from birth. The experiment timeline is shown in Fig 1E.

**4.1.2. Animal preparation.** All in vivo experiments were performed under sedation. The animals were induced into deep anesthesia in a custom box with a flow of 5% isoflurane (Vetflurane, Virbac, France) mixed with oxygen-enriched medical air for approximately 2 min. Once sedated, the animals were moved to a custom MRI animal bed (Bruker Biospin, Karlsruhe, Germany) and maintained under approximately 2.5% to 3.5% isoflurane while being prepared for imaging. The animals were placed 2.5 cm from a screen, where the stimuli were projected, and eye drops (Bepanthen, Bayer, Leverkusen, Germany) were applied to prevent the eyes from drying during anesthesia. Approximately 5 min after the isoflurane induction, a bolus (0.05 mg/kg) of medetomidine (Dormilan, Vetpharma Animal Health, Spain) consisting of a 1 mg/ml solution diluted 1:10 in saline was administered subcutaneously. Ten to 18 min after the bolus, a constant infusion of 0.1 mg/kg/h of medetomidine, delivered via a syringe pump (GenieTouch, Kent Scientific, Torrington, Connecticut, United States of America), was started. During the period between the bolus and the beginning of the constant infusion, isoflurane was progressively reduced until reaching 0%.

Temperature and respiration rate were continuously monitored via a rectal optic fiber temperature probe and a respiration sensor (Model 1025, SAM-PC monitor, SA Instruments, USA), respectively, and remained constant throughout the experiment. Each MRI session lasted between 2 h 30 and 3 h. At the end of each MRI session, to revert the sedation, 2.0 mg/kg of atipamezole (5 mg/ml solution diluted 1:10 in saline) (Antisedan, Vetpharma Animal Health, Spain) was injected subcutaneously at the same volume of the initial bolus.

**4.1.3. MRI acquisition.** All the MRI scans were performed using a 9.4T Bruker BioSpin MRI scanner (Bruker, Karlsruhe, Germany) operating at a $^1$H frequency of 400.13 MHz and equipped with an AVANCE III HD console and a gradient system capable of producing up to 660 mT/m isotropically. An 86 mm volume quadrature resonator was used for transmittance and a 20 mm loop surface coil was used for reception. The software running on this scanner was ParaVision 6.0.1.

After placing the animal in the scanner bed, localizer scans were performed to ensure that the animal was correctly positioned and routine adjustments were performed. B0 maps were acquired. A high-definition anatomical $T_2$-weighted Rapid Acquisition with Refocused Echoes (RARE) sequence (TE/TR = 13.3/2,000 ms, RARE factor = 5, FOV = $20 \times 16$ mm$^2$, in-plane resolution = $80 \times 80$ μm$^2$, slice thickness = 500 μm, $t_{acq}$ = 1 min 18 s) was acquired for accurate referencing. Functional scans were acquired using a gradient-echo echo-planar imaging (GE-EPI) sequence (TE/TR = 16.7/1,500 ms, FOV = $20.5 \times 15$ mm$^2$, resolution = $125 \times 125$ μm$^2$, slice thickness = 650 μm, 12 slices covering the visual pathway, flip angle = 15˚). Importantly, the fMRI scans were started approximately 30 min after the

isoflurane was removed from the breathing air to avoid the potentially confounding effects of isoflurane [80].

We performed 2 types of visual stimulus: retinotopy mapping and spatial frequency tuning. The animals underwent a total of 6 runs of the retinotopic stimulus, each run taking 7 min and 39 s to acquire (306 repetitions), and 2 runs of spatial frequency tuning experiment, each lasting 12 min and 15 s (490 repetitions).

## 4.2. Visual stimulus delivery setup and paradigm

**4.2.1. Setup.** The complex visual stimuli necessary for retinotopic mapping, insofar never achieved in rodent fMRI, were generated outside the scanner and back-projected with an Epson EH-TW7000 projector onto a semitransparent screen positioned 2.5 cm from the animals eyes (Fig 1A and 1B). The projector was located in the scanner room outside the fringe field. An acrylic mirror of $30 \times 30$ cm$^2$ was positioned approximately 2 meters from the projector and approximately 2.6 meters from the animal. The mirror was angled at 45˚ so that the light coming from the projector was reflected with a 90˚ angle towards the scanner bore. The light was focused in a semi-circular screen with 4 cm radius, resulting in a field of view of approximately 116˚ of visual angle. To avoid occlusion of the field of view, the amplifier was placed behind the animal. A scheme of the setup is shown on Fig 1A and Fig A in S1 Text shows pictures of the visual setup mounted to the MRI animal cradle. The animals viewed the stimulus binocularly. Using this setup, 2 sets of stimuli were presented: a retinotopy stimulus that allows us to derive the pRF parameters, and a spatial frequency tuning stimulus. Visual stimuli were created using MATLAB (Mathworks, Natick, Massachusetts, USA) and the Psychtoolbox. An Arduino MEGA260 receiving triggers from the MRI scanner was used to control stimuli timings.

**4.2.2. Retinotopy.** The visual stimuli consisted of a luminance contrast-inverting checkerboard drifting bar [30]. The bar aperture was composed by alternating rows of high-contrast luminance checks moved in 8 different directions (4 bar orientations: horizontal, vertical, and the 2 diagonal orientations, with 2 opposite drift directions for each orientation, Fig 1C). The bar moved across the screen in 16 equally spaced steps, each lasting 1 TR. The bar contrast, width, and spatial frequency were 50%, approximately 14.5˚, and approximately 0.2 cycles per degree of visual angle (cpd), respectively. The retinotopic stimulus consisted of 4 stimulation blocks. At each stimulation block, the bar moved across the entire screen during 24 s (swiping the visual field in the horizontal or vertical directions) and across half of the screen for 12 s (swiping half of the visual field diagonally), followed by a blank full screen stimulus at mean luminance for 45 s, Fig 1C. A single retinotopic mapping run consisted of 246 functional images (60 pre-scan images were deliberately planned to be discarded due to coil heating).

**4.2.3. Spatial frequency tuning.** The stimulus consisted of a block design on/off task, with a baseline of 45 s which consisted of a black screen, and an activation task of 15 s (Fig 1D), in a total of 10 stimulation blocks. The activation stimulus consisted of vertical sinusoidal gratings of multiple spatial frequencies: 0.01, 0.02, 0.03, 0.06, 0.08, 0.1, 0.16, 0.25, 0.36, 0.5 cpd moving left to right at 5 Hz. The SF stimulation blocks were randomized. The grating contrast was 100%. A single retinotopic mapping run consisted of 430 functional images (60 pre-scan images were deliberately planned to be discarded due to coil heating).

## 4.3. Hypercapnia

A hypercapnia experiment was performed to determine the influence of the vascular component in putative changes in activation, retinotopic maps, and SF tuning curves, and to

disentangle between the neural and vascular components of the BOLD responses in multiple regions of the visual pathway. To achieve this, $N = 10$ additional animals ($N = 6$ females, 12 to 28 weeks old, median weight 340 g; range 285 to 470 g) underwent the hypercapnia condition (VD, $N = 5$, HC, $N = 5$). The animals performed the hypercapnia condition at t = 0. The hypercapnia paradigm consisted of 90 s ventilation with medical air followed by a manual switch to a hypercapnia state with 6.5% $CO_2$ for 90 s. In the end of the hypercapnia period, the $CO_2$ was switched off, the animals resumed breathing medical air, and data kept being acquired for 1.5 additional minutes. In between hypercapnia "runs," the animals rested for 2 min.

## 4.4. Data analysis

**4.4.1. Preprocessing.** The images were first converted to Nifti. Then, outlier correction was performed (time points whose signal intensity was 3 times higher or lower than the standard deviation of the entire time course were replaced by the mean of the 3 antecedent and 3 subsequent time points). Simultaneous motion and slice timing correction were performed using Nipy's SpaceTimeRealign function [102]. The brain extraction was done using AFNI function Automask applied to a bias field corrected (ANTs) mean functional image. The skull stripped images were inspected and upon visual inspection a further mask was manually drawn when needed using a home-written script. The skull stripped images then underwent co-registration and normalization to an atlas [103]. The co-registration alignment was performed by calculating the transform matrix that aligns the mean functional image of each run to the anatomical image, and normalization was performed by calculating the transform matrix that aligns each anatomical image with the atlas template. These 2 sets of matrices were then applied to all the runs per animal. Co-registration and normalization were performed in ANTs and, when necessary, manually adjusted using ITK SNAP. Following normalization, the voxels' signals were detrended using a polynomial function of factor 2 fitted to the resting periods and spatially smoothed (FWHM = 0.15 mm$^2$). All preprocessing steps were performed using a home-written Python pipeline using Nypype.

**4.4.2. ROI analysis.** Five ROIs were defined according to the SIGMA atlas [103] and manually adjusted per animal. These ROIs comprehended different visual pathway structures such as the binocular primary VC, the LGN, and the superior layer of the SC. SC is divided into superficial and deep layers. While the superficial layers have a purely visual sensory role, deep layers have both multisensory (auditory, tactile, or visual) and motor functions [104]. For this reason, the SC ROI concerns only the SC superficial gray layers. Note that the SC ROI overlays with the stronger activated region of the SC (Fig B in S1 Text). Similarly, the visual cortex ROI was defined as the primary visual cortex (V1). In particular in response to the spatial frequency tuning stimuli the response in the VC is spatially broader than V1, which corresponds to the activation of higher visual order (V2) areas (Fig B in S1 Text).

The detrended time series of the voxels comprising each ROI were converted into percentage signal change (PSC) and averaged across epochs, runs, and animals providing the averaged response within each region per acquisition time point. The 25%, 50%, and 75% percentiles per ROI were also calculated (Fig 4 and Figs B, C, D, and I in S1 Text).

**4.4.3. Retinotopic mapping analysis.** Retinotopic mapping analysis was performed using both conventional pRF mapping [30] and micro-probing [36]. In brief, these methods model the population of neurons measured within a voxel as a 2D Gaussian, where the center corresponds to the pRF's position and the width to its size.

**4.4.4. Conventional pRF mapping.** In the conventional method, a 2D Gaussian model $n$ $(x,y)$ was fitted with parameters: center ($x_0$, $y_0$) and size ($\sigma$—width of the Gaussian), for each

voxel.

$$n(x, y) = e^{\frac{(x-x_0)^2 + (y-y_0)^2}{-2\sigma^2}} \tag{1}$$

The predicted response of a voxel $p(t)$ to the stimulus was then calculated as the overlap between the stimulus mask (binary image of the stimulus aperture over time: $s(x,y,t)$) at each time point and the neural model $n(x,y)$.

$$p(t) = \sum_{x,y} s(x, y, t) * n(x, y) \tag{2}$$

Subsequently, the delay in hemodynamic response was accounted for by convolving the predicted time courses with a hemodynamic response function (HRF) model consisting of a double-gamma function with a peak at 1.4 s. Finally, the pRF model parameters were adjusted for each cortical location to minimize the difference between the prediction and the measured BOLD data. The best fitting parameters are the output of the analysis.

The pRF properties estimation was performed using a home-written script. The data was thresholded by retaining the pRF models that explained at least 5% of the variance. The number of voxels excluded due to the VE threshold is shown in Table E in S1 Text.

**4.4.5. Micro-probing.** Micro-probing applies large numbers of "micro-probes," 2D Gaussians with a narrow standard deviation, to sample the entire stimulus space and create high-resolution probe maps. The number of micro-probes included, 10,000, was calculated based on the trade-off between achieving a good coverage of the visual field and the time to compute a probe map. Like the conventional pRF approach, these micro-probes sample the aggregate response of neuronal subpopulations, but they do so at a much higher spatial resolution. Consequently, for each voxel, the micro-probing generates an RF profile representing the density and VE for all the probes.

**4.4.6. Quantification of the retinotopic organization of the visual pathway.** The quantification of the topographical organization of the visual phase was done based on the pRF phase across the gradient axis. First, the phase gradient axis was defined for each ROI as shown in Fig F in S1 Text. Second, the pRF phase estimates were projected on the phase gradient axis and then averaged across the gradient axis distance. Third, a linear function was fitted between the averaged phase estimates and the cortical distance according to the gradient axis. The relation (slope of the linear function) between the phase and cortical distance was calculated for HC and VD animals for all the scanning sessions. Importantly, we only included in this analysis the animals that completed all 4 scanning sessions ($N = 6$). Fourth, a statistical comparison between the slopes obtained for HC and VD for all the scanning sessions was performed. ANOVA with post hoc analysis Bonferroni was used to correct for multiple comparisons (brain area and scanning sessions).

**4.4.7. Spatial frequency analysis.** The optimal SF per voxel was determined. First, we calculated the maximum BOLD modulation during the activation period of each individual activation block. The SF that elicited the strongest BOLD response was considered the optimal SF for that specific voxel.

The ROI specific tuning curves were obtained by calculating the optimal SF on the averaged ROI signal during the activation period of each individual activation block. Then, we plotted the BOLD magnitude (i.e., the maximum PSC) of the activation blocks as a function of the SF. Finally, we fitted a 1D Gaussian model to the SF tuning curve:

$$spatial\ frequency_{tuning\ curve} = A \cdot e^{\frac{-(x-\mu)^2}{2\sigma^2}} \tag{3}$$

where the center ($\mu$) corresponds to the optimal SF and $\sigma$ to the broadness of the tuning curve.

**4.4.8. Statistical analysis.** All statistical analyses were performed using MATLAB (version 2016b; Mathworks, Natick, Massachusetts, USA) and Rstudio. Unless otherwise specified, after correction for multiple comparisons, a $p$-value of 0.05 or less was considered statistically significant.

For the statistical t-maps, different GLMs were fitted, targeting (1) a per-subject analysis and (2) a group-level analysis. In all cases, each session from every animal was regressed with their respective realignment parameters and a double-gamma HRF (described in section 4.4.4) convolved with the stimulation paradigm (contrast obtained using on and off blocks).

The t-values associated with the activation contrast were then mapped voxelwise. The maps were FDR corrected for multiple comparisons using a $p$-value of 0.001 and minimum cluster size of 20 voxels.

Statistical comparisons of the BOLD magnitude, pRF properties (size and phase slope), and SF tuning curves were performed using ANOVA with post hoc analysis Bonferroni corrected for multiple comparisons (brain area and scanning sessions).

## Supporting information

**S1 Text. Fig A in S1 Text. Photographs of the visual setup mounted to an MRI animal cradle.** Animal life-support equipment: temperature and respiration sensors, subcutaneous catheter, and homeothermic water blanket were connected to the cradle. The surface coil was placed on the head of the rats and the preamplifier was placed behind the animal. The visual stimulus was projected to a mirror that reflected the image to the screen placed in front of the animal eyes. **Fig B in S1 Text. Retinotopic and SF tuning visual stimuli result in robust BOLD signals confined to the visual pathway in HC at t = 0.** A, B, and E: Percentage of BOLD signal change (PSC) of the ROIs defined in D averaged across animals, runs (B), and cycles (A, E), upon retinotopic (A, B) and SF tuning (E) visual stimulation. The colored areas correspond to the 95% confidence interval and the gray area to the stimulation period. C and F: GLM functional maps obtained after retinotopic (C) and SF tuning (F) visual stimulation. The maps are FDR corrected using a $p$-value of 0.001 and minimum cluster size of 20 voxels. The ROIs defined based on the SIGMA atlas are overlaid on the functional maps. D: Anatomical images with the delineation of the ROIs. The data underlying this figure can be found here: doi:10.18112/openneuro.ds004509.v1.0.0. **Fig C in S1 Text. Differential responses between VD animals and HC driven by the SF tuning stimulus.** A: Raw fMRI images with the ROIs (LGN, SC, and VC) overlaid. B, E, H, K: fMRI activation patterns of t-contrast maps obtained for HC and VD animals at t = 0, t = 7d, t = 17d, and t = 27d, respectively. The GLM maps are FDR corrected using a $p$-value of 0.001 and minimum cluster size of 20 voxels. C, F, I, L: PSC of the LGN, SC and VC for the HC and VD animals at t = 0, t = 7d, t = 17d, and t = 27d, respectively. The gray area represents the stimulation period. D, G, J, M: Violin plot of the amplitude of the BOLD response of VD and HC during the total duration of the activation period obtained with the SF tuning stimulus (right) at t = 0, t = 7d, t = 17d, and t = 27d, respectively. The white dot represents the mean, and the gray bar represents the 25% and 75% percentiles. The blue, yellow, and red colors represent the VC, LGN, and SC, respectively. The *** represents a $p$-value <0.001, ** $p$-value <0.01, and * $p$-value <0.05. The data underlying this figure can be found here: doi:10.18112/openneuro.ds004509.v1.0.0. **Fig D in S1 Text. Differential responses between VD animals and HC driven by the retinotopic stimulus.** A: Raw fMRI images with the ROIs (LGN, SC, and VC) overlaid. B, E, H, K: fMRI activation patterns of t-contrast maps obtained for HC and VD animals at t = 0, t = 7d, t = 17d, and t = 27d, respectively. The GLM maps are FDR corrected using a $p$-value of 0.001 and minimum cluster size of 20 voxels. C, F, I, L: PSC of the LGN, SC, and VC for the HC and VD animals at t = 0, t = 7d, t = 17d, and t = 27d, respectively. The gray area represents the stimulation period. D, G, J,

M: Violin plot of the amplitude of the BOLD response of VD and HC during the total duration of the activation period obtained with the SF tuning stimulus (right) at t = 0, t = 7d, t = 17d, and t = 27d, respectively. The white dot represents the mean, and the gray bar represents the 25% and 75% percentiles. The blue, yellow, and red colors represent the VC, LGN, and SC, respectively. The *** represents a *p*-value <0.001, ** *p*-value <0.01, and * *p*-value <0.05. The data underlying this figure can be found here: doi:10.18112/openneuro.ds004509.v1.0.0. **Fig E. Spatial frequency selectivity across the visual pathway in HC animals at t = 0.** A: Anatomical images with the ROIs highlighted. B: Optimal spatial frequency estimated per voxel for HC, averaged across animals. C: Maximum PSC during the activation period as a function of the spatial frequency of the stimulus, calculated for HC. The error bar represents the 10% confidence interval across animals. The continuous lines represent the Gaussian model fitted to the data. The goodness of fit is shown in Table D. The orange and red bands denote the range of optimal spatial frequency values reported in the literature measured using electrophysiology for rats and mice, respectively. A compilation of 22 studies reporting on the optimal spatial frequency of the rat and mouse visual pathway can be found in Table C. The data underlying this figure can be found here: doi:10.18112/openneuro.ds004509.v1.0.0. **Fig F in S1 Text. Gradient direction for VC, LGN, and SC. Fig G in S1 Text. Quantification of the topographical organization of the visual pathway.** A: pRF profiles of 2 adjacent pRFs. B: Distance matrix of all the SC voxels. C: PRF similarity analysis of all SC voxels averaged for all the animals. The data underlying this figure can be found here: doi:10.18112/openneuro.ds004509.v1.0.0. **Fig H in S1 Text. PRF estimates in the AC, MC, and BG.** A and B: Violin plots of the variance explained calculated for visual areas (VC, LGN, and SC) and for areas not visually responsive (AC, MC, and BG) for HC and VD, respectively. C: Anatomical image with the ROIs AC, MC, and BG overlapped. D: PRF profiles obtained for HC and VD animals (left and right, respectively) for AC, MC, and BG. The data underlying this figure can be found here: doi:10.18112/openneuro.ds004509.v1.0.0. **Fig I in S1 Text. Hypercapnia experiment testing the dynamics of vascular responses.** A: Hypercapnia paradigm consisted of a manual switch, after 1.5 min of medical air, to a hypercapnic state with 6.5% $CO_2$ for 1.5 min. This was followed by a manual switch again to medical air for 1.5 min. Each run consisted in only 1 repetition of this block. B, C, and D: PSC response profile (mean ± std) obtained for VD (red) and HC (blue) for different ROIs: VC, LGN, and SC, respectively. The shaded gray area indicates the hypercapnic period. E, F and G: Normalized PSC response profile (mean ± std) obtained for VD (red) and HC (blue) for different ROIs: VC, LGN, and SC, respectively. The data underlying this figure can be found here: doi:10.18112/openneuro.ds004509.v1.0.0. **Table A in S1 Text. Statistical analysis HC vs. VD BOLD changes.** *P*-values associated with the ANOVA Bonferroni corrected for multiple comparisons (brain areas and sessions) statistical analysis of the BOLD amplitude changes between HC and VD in response to the retinotopic stimulus (Fig 4). The data underlying this table can be found here: doi:10.18112/openneuro.ds004509.v1.0.0. **Table B in S1 Text. *P*-values obtained for the pRF size changes between HC and VD, calculated using ANOVA Bonferroni corrected for multiple comparisons (brain areas and sessions).** The data underlying this table can be found here: doi:10.18112/openneuro.ds004509.v1.0.0. **Table C in S1 Text. Summary of the optimal spatial frequency measured for rat and mice across multiple studies.** The rat studies used to define the reference interval of Fig 3 are highlighted in yellow. **Table D in S1 Text. Pearson's coefficient between the maximum BOLD response to each SF and the Gaussian fit.** The data underlying this table can be found here: doi:10.18112/openneuro.ds004509.v1.0.0. **Table E in S1 Text. Fraction of voxels excluded by the variance explained threshold.** The data underlying this table can be found here: doi:10.18112/openneuro.ds004509.v1.0.0.

(DOCX)

## Acknowledgments

We would like to thank Ms. Rita Gil and Mr. Federico Severo for setting up the visual deprivation room.

## Author Contributions

**Conceptualization:** Joana Carvalho, Noam Shemesh.

**Data curation:** Joana Carvalho, Francisca F. Fernandes.

**Formal analysis:** Joana Carvalho.

**Funding acquisition:** Joana Carvalho, Noam Shemesh.

**Investigation:** Joana Carvalho, Francisca F. Fernandes, Noam Shemesh.

**Methodology:** Joana Carvalho, Francisca F. Fernandes, Noam Shemesh.

**Project administration:** Joana Carvalho, Noam Shemesh.

**Resources:** Joana Carvalho, Noam Shemesh.

**Software:** Joana Carvalho.

**Supervision:** Noam Shemesh.

**Validation:** Joana Carvalho, Noam Shemesh.

**Visualization:** Joana Carvalho, Noam Shemesh.

**Writing – original draft:** Joana Carvalho, Noam Shemesh.

**Writing – review & editing:** Joana Carvalho, Francisca F. Fernandes, Noam Shemesh.

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
