## [Editor Report · Decision Letter 0]

13 Feb 2023

Dear Dr Shemesh, 

Thank you for clarifying the methods on your manuscript entitled "Extensive topographic remapping and functional sharpening in the adult rat visual pathway upon first visual experience" for consideration as a Research Article by PLOS Biology. I've now had a chance to discuss this with our Academic Editor and we agree that the study is suitable for external peer review at PLOS Biology. 

IMPORTANT: before we can send your manuscript to reviewers, we need you to complete your submission by providing the metadata that is required for full assessment. To this end, please login to Editorial Manager where you will find the paper in the 'Submissions Needing Revisions' folder on your homepage. Please click 'Revise Submission' from the Action Links and complete all additional questions in the submission questionnaire. At this point you will also be offered the opportunity to suggest suitable reviewers, without conflicts of interest, who can cover the various aspects of your work. While this is not required, the editors do find such suggestions helpful. You will also be offered the opportunity to exclude up to three reviewers.

Once your full submission is complete, your paper will undergo a series of checks in preparation for peer review. After your manuscript has passed the checks it will be sent out for review. To provide the metadata for your submission, please Login to Editorial Manager (https://www.editorialmanager.com/pbiology) within two working days, i.e. by Feb 15 2023 11:59PM.

Feel free to email us at plosbiology@plos.org if you have any queries relating to your submission. I also apologize for the delay in getting back to you with this decision. It took a few days for the Academic Editor to find time to look the submission over and get back to me.

Kind regards,

Kris

Kris Dickson, Ph.D., (she/her)

Neurosciences Senior Editor/Section Manager

PLOS Biology

kdickson@plos.org

---

## [Decision Letter · Decision Letter 1]

24 Apr 2023

Dear Dr Shemesh,

Thank you for your patience while your manuscript "Extensive topographic remapping and functional sharpening in the adult rat visual pathway upon first visual experience" was peer-reviewed at PLOS Biology. It has now been evaluated by the PLOS Biology editors, an Academic Editor with relevant expertise, and by several independent reviewers. In light of the reviews, which you will find at the end of this email, we would like to invite you to revise the work to address the reviewers' reports. As you will see, the reviewers find the study interesting and generally well done, but raise a number of comments and questions that we think should be thoroughly addressed. 

Given the extent of revision needed, we cannot make a decision about publication until we have seen the revised manuscript and your response to the reviewers' comments. Your revised manuscript is likely to be sent for further evaluation by all or a subset of the reviewers.

**IMPORTANT - SUBMITTING YOUR REVISION**

*Re-submission Checklist*

*Published Peer Review*

*PLOS Data Policy*

*Blot and Gel Data Policy*

Sincerely,

Luke

Lucas Smith, Ph.D.

Associate Editor

PLOS Biology

lsmith@plos.org

REVIEWS:

Reviewer #1 (Alessandro Gozzi - note, Reviewer 1 has signed this review) : In this manuscript Carvalho and colleagues non-invasively investigated sensory remapping in the adult brain using functional magnetic resonance imaging (fMRI). Using a novel set-up allowing delivery of patterned visual stimuli anesthetized rats inside an MRI scanner, the authors succeeded for the first time in mapping receptive fields (RFs) and spatial frequency tuning curves using fMRI, hence expanding the experimental repertoire available to investigate these phenomena beyond canonical (and invasive) investigational paradigms. The authors then moved on to intriguingly track RF dynamics in visually deprived rats, revealing that light exposure can progressively promote functional remapping of unrefined visual processing systems toward functional specialization.

I found this work technically superb. The possibility of tracking remapping of visual pathways non invasively over time offers exciting opportunities for both basic and translational neuroscience. 

I do not have any major criticism as most experimental confounds appear to be well controlled by these authors, who showed in this manuscript the ability to proficiently master the difficulties of fMRI in rodents. For these reasons I strongly endorse the publication of this work in Plos Biology

Below are very few minor questions/comments that mostly address my excitement in these methods, and my interest in promoting interpretability and repeatability of this measurements by others

- It would be nice to show in greater detail i.e. in a supplementary figure, a close up of the set-up when applied to the animal's head. How could the authors achieve lack of coverage of the animals eyes, owing to the need to use a (preamplified) surface coil? A few pictures showing how the animal was arranged onto the animal cradle would be useful

- Fig. S1, retinotopic stimuli: the collicular ROI chosen (D) looks slightly misplaced and potentially too small to reliably estimate the effects elicited by visual stimuli. The corresponding response is in fact spatially broader. I would probably use a larger areas capturing the broader stimulated areas (panel C)? 

Same in Fig. 4A and S2A; here the superimposed ROI seems to straddle outside border of the brain (most likely a problem of misregistration of anatomical template to EPI underlay). This alignment should probably checked and adjusted to reflect actual ROI placement for quantification.

- Analogous small misplacement in anatomical landmarking are present in Fig. 3A vs 3B. Are these functionally relevant or just a reflection of difficulties in anatomical cross-referencing of atlases?

- I found it a bit difficult to interpret information in parametric maps in many figures, but especially Fig. 5 or Fig. 7 in which most of the space is taken up by non-relevant anatomical underlays showing brain parenchima. Perhaps a different rearrangement of the panels allowing to zoom into visually relevant regions may make the encoded information more directly interpretable to readers?

- One aspect that I think should be emphasized in discussion is the direct translatability of this fMRI paradigms across-species, including humans. In this respect I found this work of extreme interest as most investigations of these phenomena in animals entail invasive method not directly accessible to human experimenters.

Reviewer #2 (Nanyin Zhang - note, Reviewer 2 has signed this review) : In this paper authors investigated the neuroplasticity along the visual pathway using fMRI in a rodent model of visual deprivation. Using a novel setup design, they mapped in detail the topographical and neuroanatomical organization of the entire rodent visual pathway. Their data show that light exposure in adulthood results in an extensive topographical remapping and functional sharpening. Understanding the dynamics of stability/plasticity balances and how they are changed by experience is critical. Therefore, this study will have high impact on the neuroscience field. I have a few questions/comments that I hope authors can help address.

1. How do researchers ensure the animal's eyes focus on the fixation point at the center of the visual field? The retinotopic mapping need to rely on this condition.

2. 'we find no organization of spatial frequency selectivity, which indicates that each area responds globally to a set of specific spatial frequencies.' Could lack of spatial frequency selectivity be due to too coarse spatial resolution of fMRI?

3. If LGN and SC are both relays of visual information, why do they have higher sensitivity to lower spatial frequencies than VC? 

4. Could VDM also cause alterations in vascular development, which can contribute to different BOLD responses between VD and HC animals.

5. I suggest authors provide some quantitative assessment on the topographical remapping of pRF positions. For example, for each ROI, can spatial correlation values of RF position maps be provided between VD and HC animals at separate time points?

Reviewer #3 (Stelios M Smirnakis - - note, Reviewer 3 has signed this review): This paper addresses a problem of general interest. It is technically sound and well written and its conclusions appear in general justified. Still there are some areas in the paper where further clarifications would be useful to the reader and would enhance confidence in the results: 

1. Figure 3 is a bit surprising in that the peak of the frequency response appears to be quite narrow, more so that one would expect from electrophysiological or psychophysical studies in the rodents. Perhaps this narrowness is an artifact of the fit or a technical issue with fMRI imaging normalization, but it would be nice to comment on it and offer an explanation. 

2. In figure 4, the number of animals listed in the methods seems is 10 but the number of points in the violin plots is greater (perhaps 20). I assume this is because slices are plotted rather than animals. Although the results look significant, it would be important to report whether statistical comparisons involve comparison across animals rather than slices and whether they are affected by multiple comparison corrrections (was the multiple comparison across the 3 areas or also time points?). 

3. In the description of figure 4 in the text there appear to be several apparent description mistakes. For example, in line 198 - "in the first 10s of visual stimulation .... the bold response in VC and LGN in the VD group was significantly higher than in HC (fig. 4C,D)" -- however in figure 4C,D the groups is labeled as DR and there does not appear to be significant difference in LGN. In line 210-11 "for all time points, the VD SC exhibited a negative bold response to the retinotopic stimulus..." but not all the mean SC responses in the figure appear to be negative. I recommend aligning the labels (VD, DR) between the text and figure and reviewing the text description more carefully to avoid inaccuracies. Also it would be good to review abbreviations and ensure they are defined during the first time of use.

4. Line 177 -- peak spatial frequency response in visual cortex I believe should be at 0.1 cpd not 1 cpd 

5. In figure 6, it would be helpful to make a statistical comparison of the phase maps across animals to quantitatively assess the significance and magnitude of the observed changes. 

6. Line 254 --" HC pRF estimates for VC LGN and SC did not significantly differ between scanning sessions" -- In figure 6A for LGN it appears that the difference between t=0 and t=7 d for HC (green) might actually be significant -- would it be possible to re-check-- Also the error bars do not appear to reflect standard deviation as stated in the legend (see for example the brown bar, LGN, t=27d). It would be good to state in the legend whether the statistical tests were done across animals and whether multiple comparisons were implemented or not. Was ANOVA used or pairwise t-tests? 

7. The example given in 6B does not appear to be the best example of the evolution of pRf sizes seen in line 6C. The color map represents variance explained but is this done by micro-probing or by conventional prf mapping? what is the variance explained that would be corresponding to noise -- ie from voxels in an area that is not visually responsive? 

8. In figure 7: is time point 17d reliable? It appears very little activity is seen in VC for this animal. Is this physiological or is it possible that it represents experimental/technical variability. It also appears the LGN, SC responses are not just responses to spatial frequency but to luminance as the baseline remains high throughout. Was the stimulus chosen isoluminant across conditions? If so what is causing the PSC to stay high across SF conditions for LGN and SC? 

9. It would be good to discuss whether the refinement of pRF responses is due to single cell receptive field refinement versus decrease of receptive field variability across neurons that belong in the same voxels. Also it would be useful to discuss more specifically why the experimental differences between the present study and reference 33 would explain how results differ.

---

## [Editor Report · Decision Letter 2]

3 Jul 2023

Dear Dr Shemesh,

Thank you for the submission of your revised Research Article "Extensive topographic remapping and functional sharpening in the adult rat visual pathway upon first visual experience" for publication in PLOS Biology. Your revised manuscript has now been assessed by the PLOS Biology editorial team, and by the Academic Editor, who has commented that the revisions are thorough and who is fully satisfied by the changes made in response to the previous reviews. Therefore, on behalf of my colleagues and the Academic Editor, Mathew Diamond, I am pleased to say that we can in principle accept your manuscript for publication, provided you address any remaining formatting and reporting issues. These will be detailed in an email you should receive within 2-3 business days from our colleagues in the journal operations team; no action is required from you until then. Please note that we will not be able to formally accept your manuscript and schedule it for publication until you have completed any requested changes.

**IMPORTANT: As you address any formatting requests to come, please also address the following editorial requests: 

1) Per journal policy, please specify the model organism studied here in the abstract. 

2) Please update the ethics statement in your methods section to include the protocol numbers for the animal care and use protocols approved by the competent institutional (Champalimaud Animal Welfare Body) and national (Direcção Geral de Alimentação e Veterinária, DGAV) authorities

3) Thank you for providing the data and code related to your manuscript as a deposition to OpenNuero (doi:10.18112/openneuro.ds004509.v1.0.0). Please update the data availability statement in our online system to reference this dataset. We also ask that you update the figure legends of your manuscript (including supplemental) to include a sentence referencing the raw data. For example, you could add the sentence "the data underlying this figure can be found here: doi:10.18112/openneuro.ds004509.v1.0.0)

PRESS

Sincerely, 

Lucas Smith, Ph.D.

Senior Editor

PLOS Biology

lsmith@plos.org